# Equivariant 3D-Conditional Diffusion Models for Molecular Linker Design

## Abstract

Fragment-based drug discovery has been an effective paradigm in early-stage drug development. An open challenge in this area is designing linkers between disconnected molecular fragments of interest to obtain chemically-relevant candidate drug molecules. In this work, we propose DiffLinker, an E(3)-equivariant 3D-conditional diffusion model for molecular linker design. Given a set of disconnected fragments, our model places missing atoms in between and designs a molecule incorporating all the initial fragments. Unlike previous approaches that are only able to connect pairs of molecular fragments, our method can link an arbitrary number of fragments. Additionally, the model automatically determines the number of atoms in the linker and its attachment points to the input fragments. We demonstrate that DiffLinker outperforms other methods on the standard datasets generating more diverse and synthetically-accessible molecules. Besides, we experimentally test our method in real-world applications, showing that it can successfully generate valid linkers conditioned on target protein pockets.

## 1 Introduction

The space of pharmacologically-relevant molecules is estimated to exceed $10^{60}$ structures (Virshup et al., 2013), and searching in that space poses significant challenges for drug design. A successful approach to reduce the size of this space is to start from *fragments*, smaller molecular compounds that usually have no more than 20 heavy (non-hydrogen) atoms. This strategy is known as fragment-based drug design (FBDD) (Erlanson et al., 2016). Given a protein pocket (a part of the target protein that employs suitable properties for binding a ligand), computationally determining fragments that interact with the pocket is a cheaper and more efficient alternative to experimental high-throughput screening methods (Erlanson et al., 2016). Once the relevant fragments have been identified and docked to the target protein, it remains to combine them into a single, connected molecule. Among various strategies such as fragment linking, merging, and growing (Lamoree & Hubbard, 2017), the former has been preferred as it allows to boost rapidly the binding energy of the target and the compound (Jencks, 1981; Hajduk et al., 1997). This work addresses the fragment linking problem.

Early computational methods for molecular linker design were based on database search and physical simulations (Sheng & Zhang, 2013), both of which are computationally intensive. Therefore, there is increasing interest for machine learning methods that can go beyond the available data and design diverse linkers more efficiently. Existing approaches are either based on syntactic pattern recognition (Yang et al., 2020) or on autoregressive models (Imrie et al., 2020; 2021; Huang et al., 2022). While the former method operates solely on SMILES (Weininger, 1988), the latter take into account 3D positions and orientations of the input fragments, as this information is essential for designing stable molecules in various application (see Appendix A.1 for details). However, these methods are not equivariant with respect to the permutation of atoms and can only combine pairs of fragments.

Linker design depends on the target protein pocket, and using this information correctly can improve the affinity of the resulting overall compound. To date, however, there is no computational method for molecular linker design that takes the pocket into account.

In this work, we introduce DiffLinker, a conditional diffusion model that generates molecular linkers for a set of input fragments represented as a 3D atomic point cloud. First, our model generates the size of the prospective linker and then samples initial linker atom types and positions from the

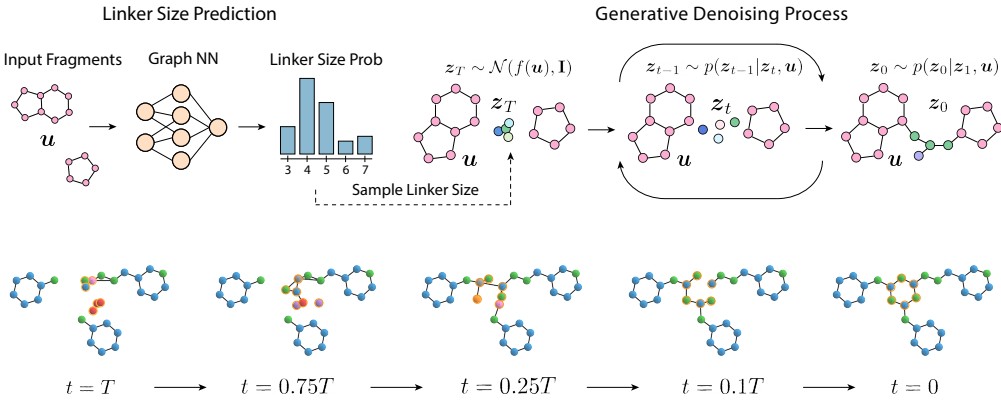

Figure 1: (Top) Overview of the molecular linker generation process. First, probabilities of linker sizes are computed for the input fragments. Next, linker atoms are sampled and denoised using our fragment-conditioned equivariant diffusion model. (Bottom) Example of the linker generation process. Linker atoms are highlighted in orange.

normal distribution. Next, the linker atom types and coordinates are iteratively updated using a neural network that is conditioned on the input fragments. Eventually, the denoised linker atoms and the input fragment atoms form a single connected molecule, as shown in Figure 1.

DiffLinker enjoys several desirable properties: it is equivariant to translations, rotations, reflections and permutations, it is not limited by the number of input fragments, it does not require information on the attachment atoms and generates linkers with no predefined size. Besides, the conditioning mechanism of DiffLinker allows to pass additional information about the surrounding protein pocket atoms, which makes the model applicable in structure-based drug design applications (Congreve et al., 2005).

We empirically show that DiffLinker is more effective than previous methods in generating chemically-relevant linkers between pairs of fragments. Our method achieves the state-of-the-art results in synthetic accessibility and drug-likeness which makes it preferable for using in drug design pipelines. Besides, DiffLinker remarkably outperforms other methods in the diversity of the generated linkers. We further propose a more challenging benchmark and show that our method is able to successfully link more than two fragments, which cannot be done by the other methods. We also demonstrate that DiffLinker can be conditioned on the target protein pocket: our model respects geometric constraints imposed by the surrounding protein atoms and generates molecules that have minimum clashes with the corresponding pockets. To the best of our knowledge, DiffLinker is the first method that is not limited by the number of input fragments and accounts the information about pockets. The overall goal of this work is to provide practitioners with an effective tool for molecular linker generation in realistic drug design scenarios.

## 2 RELATED WORK

Molecular linker design has been widely used in the fragment-based drug discovery community (Sheng & Zhang, 2013). Various de novo design methods refer to the fragment linking problem (Tschinke & Cohen, 1993; Miranker & Karplus, 1995; Roe & Kuntz, 1995; Pearlman & Murcko, 1996; Stahl et al., 2002; Thompson et al., 2008; Ichihara et al., 2011). Early fragment linking methods were based on search in the predefined libraries of linkers (Böhm, 1992; Lauri & Bartlett, 1994), genetic algorithm, tabu search (Glover, 1986) and force field optimization (Dey & Caflisch, 2008). Having been successfully used in multiple application cases (Ji et al., 2003; Silverman, 2009; Sheng & Zhang, 2011), these methods are however computationally expensive and substantially limited by the available data.

Hence, there has recently been interest in developing learning-based methods for molecular linker design. Yang et al. (2020) proposed SyntaLinker, a SMILES-based deep conditional transformer

neural network that solves a sentence completion problem (Zweig et al., 2012). This method inherits the drawbacks of SMILES, which are the absence of 3D structure and the lack of consistency (atoms that are close in the molecule can be far away in the SMILES string). Imrie et al. (2020) overcome these limitations by introducing an autoregressive model DeLinker and its extension DEVELOP (Imrie et al., 2021) that uses additional pharmacophore information. Although these methods operate on 3D molecular conformations, they use a very limited geometric information and require input on the attachment atoms of the fragments. Recently, Huang et al. (2022) have proposed another autoregressive method 3DLinker that does not require one to specify attachment points and leverages the geometric information to a much greater extent. It makes this approach more relevant for connecting docked fragments. As both DeLinker and 3DLinker are autoregressive models, they are not permutation equivariant which limits their sample efficiency and ability to scale to large molecules (Elesedy & Zaidi, 2021; Rath & Condurache, 2022). Besides, these methods are capable of connecting only pairs of fragments and cannot be easily extended to larger sets of fragments.

Outside of the linker design problem, several recent works proposed denoising diffusion models for molecular data in 3D. Conformer generation methods GeoDiff (Xu et al., 2022) and ConfGF (Shi et al., 2021) condition the model on the adjacency matrix of the molecular graph. Since they have access to the connectivity information, they can compute torsion angles between atoms and optimize them (Jing et al., 2022). Equivariant Diffusion Model (EDM) (Hoogeboom et al., 2022) generates 3D molecules from scratch and is able to be conditioned on the predefined scalar properties. Another diffusion model has been recently proposed for designing protein scaffolds given protein motifs (Trippe et al., 2022). Having the conditional sampling procedure, this model is however trained in an unconditional setup. Finally, Luo et al. (2022) proposed a model for antibody design which combines discrete diffusion for the molecular graphs and continuous diffusion on the 3D coordinates. The conditioning mechanism proposed in this work is the closest to ours, but their model is targeted at generating chains of amino acids rather than atomic point clouds.

## 3 PRELIMINARIES

### 3.1 DIFFUSION MODELS

Diffusion models (Sohl-Dickstein et al., 2015) are a class of generative methods that consist of a *diffusion process*, which progressively distorts a data point mapping it to a noise, and a *generative denoising process* which approximates the reverse of the diffusion process.

The diffusion process iteratively adds noise to a data point $\boldsymbol{x}$ in order to progressively transform it into the Gaussian noise. At a time step $t = 0, \ldots, T$, the conditional distribution of the intermediate data state $\boldsymbol{z}_t$ given the previous state is defined by the multivariate normal distribution,

$$q(\boldsymbol{z}_t | \boldsymbol{z}_{t-1}) = \mathcal{N}(\boldsymbol{z}_t; \overline{\alpha}_t \boldsymbol{z}_{t-1}, \overline{\sigma}_t^2 \boldsymbol{I}), \tag{1}$$

where $\overline{\alpha}_t \in \mathbb{R}^+$ controls how much signal is retained and $\overline{\sigma}_t \in \mathbb{R}^+$ controls how much noise is added. By hypothesis, the full transition model is Markov, so that it can be written:

$$q(\boldsymbol{z}_0, \boldsymbol{z}_1, \ldots, \boldsymbol{z}_T | \boldsymbol{x}) = q(\boldsymbol{z}_0 | \boldsymbol{x}) \prod_{t=1}^{T} q(\boldsymbol{z}_t | \boldsymbol{z}_{t-1}). \tag{2}$$

Since the distribution $q$ is normal, a simple formula for the distribution of $\boldsymbol{z}_t$ given $\boldsymbol{x}$ can be derived:

$$q(\boldsymbol{z}_t | \boldsymbol{x}) = \mathcal{N}(\boldsymbol{z}_t | \alpha_t \boldsymbol{x}, \sigma_t^2 \boldsymbol{I}), \tag{3}$$

where $\overline{\alpha}_t = \alpha_t / \alpha_{t-1}$ and $\overline{\sigma}_t^2 = \sigma_t^2 - \overline{\alpha}_t^2 \sigma_{t-1}^2$. This closed-form expression shows that noise does not need to be added iteratively to $\boldsymbol{x}$ in order to achieve an intermediate state $\boldsymbol{z}_t$.

Another key property of Gaussian noise is that the reverse of the diffusion process, the *true denoising process*, also admits a closed-form solution when conditioned on $\boldsymbol{x}$:

$$q(\boldsymbol{z}_{t-1} | \boldsymbol{x}, \boldsymbol{z}_t) = \mathcal{N}(\boldsymbol{z}_{t-1}; \boldsymbol{\mu}_t(\boldsymbol{x}, \boldsymbol{z}_t), \varsigma_t^2 \boldsymbol{I}), \tag{4}$$

where distribution parameters can be obtained analytically:

$$\boldsymbol{\mu}_t(\boldsymbol{x}, \boldsymbol{z}_t) = \frac{\overline{\alpha}_t \sigma_{t-1}^2}{\sigma_t^2} \boldsymbol{z}_t + \frac{\alpha_s \overline{\sigma}_t^2}{\sigma_t^2} \boldsymbol{x} \quad \text{and} \quad \varsigma_t = \frac{\overline{\sigma}_t \sigma_{t-1}}{\sigma_t}. \tag{5}$$

This formula simply describes that if a diffusion trajectory starts at $x$ and ends at $z_T$, then the expected value of any intermediate state is an interpolation between $x$ and $z_T$.

The second component of a diffusion model is the generative denoising process that learns to invert this trajectory having the data point $x$ unknown. The generative transition distribution is defined as:

$$p(z_{t-1}|z_t) = q(z_{t-1}|\hat{x}, z_t), \tag{6}$$

where $\hat{x}$ is an approximation of the data point $x$ computed by a neural network $\varphi$. Ho et al. (2020) have empirically shown that it works better to predict the Gaussian noise $\hat{\epsilon}_t = \varphi(z_t, t)$ instead, and then estimate the data point $\hat{x}$ using Equation (3):

$$\hat{x} = (1/\alpha_t)z_t - (\sigma_t/\alpha_t)\hat{\epsilon}_t. \tag{7}$$

The neural network is trained to maximize an evidence lower bound to the likelihood of the data under the model. Up to a prefactor that depends on $t$, this objective is equivalent to the mean squared error between predicted and true noise (Ho et al., 2020; Kingma et al., 2021). We therefore use the simplified objective $\mathcal{L}(t) = ||\epsilon - \hat{\epsilon}_t||^2$ that can be optimized by mini-batch gradient descent using an estimator $\mathbb{E}_{t \sim \mathcal{U}(0,\dots,T)}[T \cdot \mathcal{L}(t)]$.

Finally, once the network is trained, it can be used to sample new data points. For this purpose, one first samples the Gaussian noise: $z_T \sim \mathcal{N}(0, I)$. Then, for $t = T, \dots, 1$, one should iteratively sample $z_{t-1} \sim p(z_{t-1}|z_t)$ and finally sample $x \sim p(x|z_0)$.

### 3.2 Diffusion for Molecules

**Molecule Representation**  We consider now diffusion models that can operate on molecules represented as 3D atomic point clouds. A data point $x$, which is an attributed point cloud consisting of $M$ atoms, is represented by atom coordinates $r = (r_1, \dots, r_M) \in \mathbb{R}^{M \times 3}$ and the corresponding feature vectors $h = (h_1, \dots, h_M) \in \mathbb{R}^{M \times \text{nf}}$ which are one-hot encoded atom types. We will therefore denote point cloud $x$ as a tuple $x = [r, h]$.

**Categorical Features**  Along with continuous atom coordinates, a molecular diffusion model has to operate on atom types that are discrete variables. While categorical diffusion models do exist (Hoogeboom et al., 2021; Austin et al., 2021), we follow a simpler strategy (Hoogeboom et al., 2022) that is based on lifting the atom types to a continuous space: we consider a one-hot encoding of the discrete variables, and add Gaussian noise on top of it. In the end of the denoising process, once $z_0$ is sampled, the continuous values corresponding to the atom types should be converted back to discrete values. We consider the argmax over the different categories and include it in the final transition from $z_0$ to $x$. For more details on the structure of the final transition distribution $p(x|z_0)$ and likelihood computation in this setting, we refer the reader to (Hoogeboom et al., 2022).

**Equivariance**  Processing 3D molecules requires operations that respect data symmetries. In this work, we consider the Euclidean group E(3) that comprises translations, rotations and reflections of $\mathbb{R}^3$ and the orthogonal group O(3) that includes rotations and reflections of $\mathbb{R}^3$. A function $f : \mathbb{R}^3 \to \mathbb{R}^3$ is E(3)-equivariant if $f(Rx + t) = Rf(x) + t$ for any orthogonal matrix $R \in \mathbb{R}^{3 \times 3}$, $\det R = \pm 1$, for any translation vector $t \in \mathbb{R}^3$ and for any $x \in \mathbb{R}^3$. Note that for simplicity we use notation $Rx = (Rx_1, \dots, Rx_M)^\top$. A conditional distribution $p(x|y)$ is E(3)-equivariant if $p(Rx + t|Ry + t) = p(x|y)$ for any $x, y \in \mathbb{R}^3$. A function $f$ and a distribution $p$ are O(3)-equivariant if $f(Rx) = Rf(x)$ and $p(Rx|Ry) = p(x|y)$ respectively. We call the function $f$ translation invariant if $f(x + t) = f(x)$.

## 4 DiffLinker: Equivariant 3D-conditional Diffusion Model for Molecular Linker Design

In this section, we introduce DiffLinker, a new E(3)-equivariant diffusion model for generating molecular linkers conditioned on 3D fragments. We formulate equivariance requirements for the underlying denoising distributions in Section 4.1 and propose an appropriate learnable dynamic function in Section 4.2. We discuss the strategy of sampling the size of a linker and conditioning on protein pockets in Sections 4.3 and 4.4 respectively. The full linker generation workflow is schematically represented in Figure 1 and the overview of DiffLinker training and sampling procedures is provided in Algorithms 1 and 2 correspondingly.

### 4.1 EQUIVARIANT 3D-CONDITIONAL DIFFUSION MODEL

Unlike other diffusion models for molecule generation (Hoogeboom et al., 2022; Xu et al., 2022), our method is conditioned on three-dimensional data. More specifically, we assume that each point cloud $x$ has a corresponding *context* $u$, which is another point cloud consisting of all input fragments and (optionally) protein pocket atoms that remain unchanged throughout the diffusion and denoising processes, as shown in Figure 1. Hence, we consider the generative process from Equation (6) to operate on point cloud $x$ while being conditioned on the fixed corresponding context:

$$p(z_{t-1}|z_t, u) = q(z_{t-1}|\hat{x}, z_t), \quad \text{where} \quad \hat{x} = (1/\alpha_t)z_t - (\sigma_t/\alpha_t)\varphi(z_t, u, t). \quad (8)$$

The presence of a 3D context puts additional requirements on the generative process as it should be equivariant to its transformations.

**Proposition 1** *Consider a prior noise distribution $p(z_T|u) = \mathcal{N}(z_T; f(u), I)$ and transition distributions $p(z_{t-1}|z_t, u) = q(z_{t-1}|\hat{x}, z_t)$, where $q$ is an isotropic Gaussian, $f : \mathbb{R}^{M \times 3} \to \mathbb{R}^3$ is a function operating on 3D point clouds, and $\hat{x}$ is an approximation computed by the neural network $\varphi$ according to Equation (8). Let the conditional denoising probabilistic model $p$ be a Markov chain defined as*

$$p(z_0, z_1, \ldots, z_T|u) = p(z_T|u) \prod_{t=1}^{T} p(z_{t-1}|z_t, u). \quad (9)$$

*If $f$ is O(3)-equivariant and $\varphi$ is equivariant to joint O(3)-transformations of $z_t$ and $u$, then $p(z_0|u)$ is O(3)-equivariant.*

The choice of the function $f$ highly depends on the problem being solved and the available priors. In our experiments, we consider two cases. First, following (Imrie et al., 2020), we make use of the information about atoms that should be connected by the linker. We call these atoms *anchors* and define $f(u)$ as the anchors' center of mass. However, in real-world scenarios it is unlikely to know which atoms should be the anchors. In this case we define $f(u)$ as the center of mass of the whole context $u$.

We note that the probabilistic model $p$ is not equivariant to translations, as shown in Appendix A.3. To overcome this issue and make the generative process independent of translations, we construct the network $\varphi$ to be additionally translation invariant. Then, instead of sampling the initial noise from $\mathcal{N}(f(u), I)$ we center the data at $f(u)$ and sample from $\mathcal{N}(0, I)$.

### 4.2 EQUIVARIANT GRAPH NEURAL NETWORK

The learnable function $\varphi$ that models the dynamics of the diffusion model is implemented as a modified E(3)-equivariant graph neural network (EGNN) (Satorras et al., 2021). Its input is the noisy version of the linker $z_t$ at time $t$ and the context $u$. These two parts are modeled as a single fully-connected graph where nodes are represented by coordinates $r$ and feature vectors $h$ that include atom types, time $t$, fragment flags and (optionally) anchor flags. The predicted noise $\hat{\epsilon}$ includes coordinate and feature components: $\hat{\epsilon} = [\hat{\epsilon}^r, \hat{\epsilon}^h]$. In order to make function $\varphi$ invariant to translations, we subtract the initial coordinates from the coordinate component of the predicted noise following Hoogeboom et al. (2022):

$$\hat{\epsilon} = [\hat{\epsilon}^r, \hat{\epsilon}^h] = \varphi(z_t, u, t) = \text{EGNN}(z_t, u, t) - [z_t^r, 0]. \quad (10)$$

EGNN consists of the composition of Equivariant Graph Convolutional Layers $r^{l+1}, h^{l+1} = \text{EGCL}[r^l, h^l]$ which are defined as follows:

$$m_{ij} = \phi_e(h_i^l, h_j^l, d_{ij}^2), \quad h_i^{l+1} = \phi_h(h_i^l, \sum_{j \neq i} m_{ij}), \quad r_i^{l+1} = r_i^l + \phi_{vel}(r^l, h^l, i), \quad (11)$$

where $d_{ij} = \|r_i^l - r_j^l\|$ and $\phi_e$, $\phi_h$ are learnable functions parametrized by fully connected neural networks (see Appendix A.5 for details).

The latter update for the node coordinates is computed by the learnable function $\phi_{vel}$. Note that our graph includes both a noisy linker $z_t$ and a fixed context $u$, and $\varphi$ is intended to predict the noise

that should be subtracted from the coordinates and features of $z_t$. Therefore, it is natural to keep the context coordinates unchanged when computing dynamics and to apply non-zero displacements only to the linker part at each EGCL step. Hence, we model the node displacements as follows,

$$\phi_{vel}(\boldsymbol{r}^l, \boldsymbol{h}^l, i) = \begin{cases} \sum_{j \neq i} \frac{\boldsymbol{r}_i^l - \boldsymbol{r}_j^l}{d_{ij}+1} \phi_r(\boldsymbol{h}_i^l, \boldsymbol{h}_j^l, d_{ij}^2) & \text{if node } i \text{ belongs to the point cloud } \boldsymbol{z}_t, \\ 0 & \text{if node } i \text{ belongs to the context } \boldsymbol{u}, \end{cases} \quad (12)$$

where $\phi_r$ is a learnable function parametrized by a fully connected neural network.

The equivariance of convolutional layers is achieved by construction. The messages $\phi_e$ and the node updates $\phi_h$ depend only on scalar node features and distances between nodes that are E(3)-invariant. Coordinate updates $\phi_{vel}$ additionally depend linearly on the difference between coordinate vectors $\boldsymbol{r}_i^l$ and $\boldsymbol{r}_j^l$, which makes them E(3)-equivariant.

After the sequence of EGCLs is applied, we have an updated graph with new node coordinates $\hat{\boldsymbol{r}} = [\boldsymbol{u}^r, \hat{\boldsymbol{z}}_t^r]$ and new node features $\hat{\boldsymbol{h}} = [\hat{\boldsymbol{u}}^h, \hat{\boldsymbol{z}}_t^h]$. Since we are interested only in the linker-related part, we discard the coordinates and features of context nodes and consider the tuple $[\hat{\boldsymbol{z}}_t^r, \hat{\boldsymbol{z}}_t^h]$ to be the EGNN output.

### 4.3 LINKER SIZE PREDICTION

To predict the size of the missing linker between a set of fragments, we represent fragments as a fully-connected graph with one-hot encoded atom types as node features and distances between nodes as edge features. From this, a separately trained GNN (see Appendix A.6 for details) produces probabilities for the linker size. Our assumption is that relative fragment positions and orientations along with atom types contain all the information essential for predicting most likely size of the prospective linker. When generating a linker, we first sample its size with the predicted probabilities from the categorical distribution over the list of linker sizes seen in the training data, as shown in Figure 1.

### 4.4 PROTEIN POCKET CONDITIONING

In real-world fragment-based drug design applications, it often occurs that fragments are selected and docked into a target protein pocket (Igashov et al., 2022). To propose a drug candidate molecule, the fragments have to be linked. When generating the linker, one should take the surrounding pocket into account and construct a linker that has no clashes with protein pocket atoms (in other words, the configuration of the linker and pocket atoms should be physically-realistic) and keeps the binding strength high. To add pocket conditioning to DiffLinker, we represent a protein pocket as an atomic point cloud and consider it as a part of the context $\boldsymbol{u}$ (cf. Section 4.1). We also extend node features with an additional binary flag marking atoms that belong to the protein pocket. Finally, as the new context point cloud contains much more atoms, we modify the joint representation of the data point $\boldsymbol{z}_t$ and the context $\boldsymbol{u}$ that are passed to the neural network $\varphi$. Instead of considering fully-connected graphs, we assign edges between nodes based on a 4 Å distance cutoff as it makes the resulting graphs less dense and counterbalances the increase in the number of nodes.

## 5 EXPERIMENTS

### 5.1 DATASETS

**ZINC** We follow Imrie et al. (2020) and consider a subset of 250,000 molecules randomly selected by Gómez-Bombarelli et al. (2018) from ZINC database (Irwin & Shoichet, 2005). First, we generate 3D conformers using RDKit (Landrum, 2013) and define a reference 3D structure for each molecule by selecting the lowest energy conformation. Then, these molecules are fragmented by enumerating all double cuts of acyclic single bonds that are not within functional groups. The resulting splits are filtered by the number of atoms in the linker and fragments, synthetic accessibility (Ertl & Schuffenhauer, 2009), ring aromaticity, and pan-assay interference compounds (PAINS) (Baell & Holloway, 2010) criteria. One molecule can therefore result in various combinations of two fragments with a linker in between. The resulting dataset is randomly split into train (438,610 examples), validation (400 examples), and test (400 examples) sets.

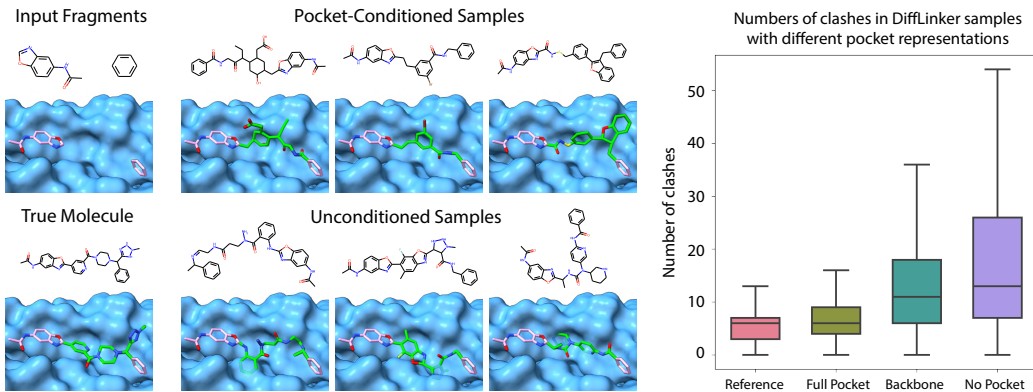

Figure 2: (Left) Examples of linkers sampled by DiffLinker conditioned on pocket atoms (top row) and unconditioned (bottom row). Linkers sampled by the unconditioned model have multiple clashes with the protein pocket. (Right) Distribution of numbers of clashes in reference test molecules and samples from three DiffLinker models differently conditioned (or unconditioned) on pocket atoms.

**CASF** Another evaluation benchmark used by Imrie et al. (2020) is taken from the CASF-2016 dataset (Su et al., 2018). In contrast to ZINC, where molecule conformers were generated computationally, CASF includes experimentally verified 3D conformations. Following the same preprocessing procedures as for the ZINC dataset, Imrie et al. (2020) obtained an additional test set of 309 examples, which we use in our work.

**GEOM** ZINC and CASF datasets used in previous works only contain pairs of fragments. However, real-world applications often require connecting more than two fragments with one or more linkers (Igashov et al., 2022). To address this case, we construct a new dataset based on GEOM molecules (Axelrod & Gómez-Bombarelli, 2022) which we decompose into three or more fragments with one or two linkers connecting them. To achieve such splits, we use RDKit implementations of two fragmentation techniques — an MMPA-based algorithm (Dossetter et al., 2013) and BRICS (Degen et al., 2008) — and combine results removing duplicates. Overall, we obtain 41,907 molecules and 285,140 fragmentations that are randomly split in train (282,602 examples), validation (1,250 examples) and test (1,288 examples) sets.

**Pockets Dataset** In order to assess the ability of DiffLinker to generate valid linkers given additional information about protein pockets, we use the protein-ligand dataset curated by Schneuing et al. (2022) from Binding MOAD (Hu et al., 2005). To define pockets, we consider amino acids that have at least one atom closer than 6 Å to any atom of the ligand. All atoms belonging to these residues constitute the pocket. We split molecules into fragments using RDKit's implementation of MMPA-based algorithm (Dossetter et al., 2013). We randomly split the resulting data into train (185,678 examples), validation (490 examples) and test (566 examples) sets taking into account Enzyme Commission (EC) numbers of the proteins.

## 5.2 EVALUATION

**Metrics** First, we report several chemical properties of the generated molecules that are especially important in drug design applications: average quantitative estimation of drug-likeness (QED) (Bickerton et al., 2012), average synthetic accessibility (SA) (Ertl & Schuffenhauer, 2009) and average number of rings in the linker. Next, following Imrie et al. (2020), we measure validity, uniqueness and novelty of the samples. We then determine if the generated linkers are consistent with the 2D filters used to produce the ZINC training set. These filters are explained in detail in Appendix A.11. In addition, we record the percentage of the original molecules that were recovered by the generation process. To compare the 3D shapes of the sampled and ground-truth molecules, we estimate the Root Mean Squared Deviation (RMSD) between the generated and real linker coordinates in the cases where true molecules are recovered. We also compute the $SC_{RDKit}$ metric that evaluates the geometric and chemical similarity between the ground-truth and generated molecules (Putta et al., 2005; Landrum et al., 2006).

Table 1: Performance metrics on ZINC, CASF and GEOM test sets. The first three metrics assess the chemical relevance of the generated molecules. The last three metrics evaluate the standard generative properties of the methods.

| | Method | QED ↑ | SA ↓ | # Rings ↑ | Valid, % | Unique, % | Novel, % |
|---|---|---|---|---|---|---|---|
| ZINC | DeLinker + ConfVAE + MMFF | 0.64 | 3.11 | 0.21 | **98.3** | 44.2 | **47.1** |
| | 3DLinker (given anchors) | 0.65 | 3.11 | 0.23 | **99.3** | 29.0 | 41.2 |
| | 3DLinker | 0.65 | 3.14 | 0.24 | 71.5 | 29.2 | 41.9 |
| | DiffLinker | **0.68** | **3.01** | 0.25 | 93.8 | 24.0 | 30.3 |
| | DiffLinker (given anchors) | **0.68** | **3.03** | 0.26 | 97.6 | 22.7 | 32.4 |
| | DiffLinker (sampled size) | 0.65 | 3.19 | **0.32** | 90.6 | **51.4** | 42.9 |
| | DiffLinker (given anchors, sampled size) | 0.65 | 3.24 | **0.36** | 94.8 | **50.9** | **47.7** |
| CASF | DeLinker + ConfVAE + MMFF | 0.35 | 4.05 | 0.35 | **95.7** | 51.6 | **55.6** |
| | DiffLinker | **0.41** | **4.00** | 0.34 | 85.3 | 40.5 | 41.8 |
| | DiffLinker (given anchors) | **0.40** | **4.03** | 0.38 | **90.2** | 37.3 | 48.4 |
| | DiffLinker (sampled size) | **0.40** | 4.06 | 0.30 | 63.7 | **60.0** | 49.3 |
| | DiffLinker (given anchors, sampled size) | **0.40** | 4.10 | **0.38** | 68.4 | **57.1** | 56.9 |
| GEOM | 3DLinker | 0.36 | 3.56 | 0.00 | 16.3 | **73.7** | — |
| | DiffLinker | **0.48** | **2.98** | 0.78 | **93.5** | 36.7 | 70.7 |
| | DiffLinker (given anchors) | **0.49** | **3.01** | 0.82 | **93.4** | 37.3 | 70.5 |
| | DiffLinker (sampled size) | 0.46 | 3.24 | 0.76 | 87.4 | 63.1 | **76.3** |
| | DiffLinker (given anchors, sampled size) | 0.47 | 3.30 | **0.84** | 88.8 | **64.4** | **76.6** |

Table 2: Metrics assessing the ability of the methods to generate molecules that are chemically and geometrically similar to the reference ones.

| | | | | | SC$_{RDKit}$ | | | |
|---|---|---|---|---|---|---|---|---|
| | Method | 2D Filters, % | Recovery, % | RMSD ↓ | > 0.7 | > 0.8 | > 0.9 | Avg ↑ |
| ZINC | DeLinker + ConfVAE + MMFF | 84.88 | 80.2 | 5.48 | 3.73 | 0.61 | 0.09 | 0.49 |
| | 3DLinker (given anchors) | 84.24 | **94.0** | **0.10** | **99.86** | **97.05** | 63.78 | 0.92 |
| | 3DLinker | 83.72 | **93.5** | **0.11** | 99.83 | 96.22 | 63.63 | 0.92 |
| | DiffLinker | **86.26** | 82.0 | 0.34 | 99.72 | 94.62 | **67.85** | **0.93** |
| | DiffLinker (given anchors) | 84.36 | 87.2 | 0.32 | **99.96** | **97.02** | **71.73** | **0.94** |
| | DiffLinker (sampled size) | **87.98** | 70.7 | 0.34 | 99.37 | 90.21 | 51.90 | 0.90 |
| | DiffLinker (given anchors, sampled size) | 84.76 | 77.5 | 0.35 | 99.67 | 95.04 | 56.35 | 0.91 |
| CASF | DeLinker + ConfVAE + MMFF | 77.25 | **52.8** | 11.89 | 1.24 | 0.19 | 0.03 | 0.39 |
| | DiffLinker | **87.73** | 42.8 | 0.44 | 92.17 | 79.62 | 50.14 | 0.84 |
| | DiffLinker (given anchors) | 82.37 | **50.2** | 0.37 | **95.07** | **89.05** | **60.63** | **0.86** |
| | DiffLinker (sampled size) | **89.27** | 40.5 | **0.34** | 92.83 | 79.12 | 43.99 | 0.82 |
| | DiffLinker (given anchors, sampled size) | 82.08 | 48.8 | **0.32** | **94.52** | **87.90** | 55.25 | **0.84** |
| GEOM | 3DLinker | **94.10** | 0.0 | — | 60.63 | 29.30 | 8.22 | 0.72 |
| | DiffLinker | 47.68 | **85.6** | 0.12 | **95.53** | **87.87** | **71.36** | **0.92** |
| | DiffLinker (given anchors) | 49.70 | **85.2** | 0.12 | **95.12** | 86.79 | 69.91 | **0.92** |
| | DiffLinker (sampled size) | 55.68 | 68.8 | **0.07** | 91.74 | 81.47 | 57.11 | 0.88 |
| | DiffLinker (given anchors, sampled size) | **57.90** | 67.9 | **0.07** | 90.95 | 79.57 | 54.78 | 0.87 |

**Baselines** We compare our method with DeLinker (Imrie et al., 2020) and 3DLinker (Huang et al., 2022) on the ZINC test set and with DeLinker on the CASF dataset. Besides, we adjust 3DLinker to connect more than two fragments (see Appendix A.7 for details) and evaluate its performane on the GEOM dataset. To obtain 3D conformations for the molecules generated by DeLinker on ZINC and CASF, we apply a pre-trained ConfVAE (Xu et al., 2021) followed by a force field relaxation procedure. For all methods including ours, we generate linkers with the ground-truth size unless explicitly noted otherwise. To obtain SMILES representations of atomic point clouds generated by our models, we utilize OpenBabel (O'Boyle et al., 2011) to compute covalent bonds between atoms. We also use OpenBabel to rebuild covalent bonds for the molecules in test sets in order to correctly compute the recovery rate, RMSD and SC$_{RDKit}$ scores for our models. In ZINC and CASF experiments, we sample 250 linkers for each input pair of fragments. For the GEOM dataset and in experiments with pocket conditioning, we sample 100 linkers for each input set of fragments.

### 5.3 RESULTS

**ZINC and CASF**  While our models have much greater flexibility and applicability in more applications as we show below, they also outperform other methods on standard benchmarks ZINC and CASF in terms of chemical relevance of the generated molecules. As shown in Table 1, molecules sampled by DiffLinker are more synthetically accessible and demonstrate higher drug-likeness, which is especially important for drug design applications. Besides, our models generate linkers containing more rings. Moreover, our molecules usually share higher chemical and geometric similarity with the reference molecules as demonstrated by the $SC_{RDKit}$ scores in Table 2. In terms of validity, our models perform on par with the other methods. Note that both autoregressive approaches employ valency rules at each generation step explicitly, while our model is shown to be able to learn these rules from the data. Remarkably, the validity of the reference molecules from CASF with covalent bonds computed by OpenBabel is 92.2% while our model generated molecules with 90.2% validity. Notably, sampling the size of the linker significantly improves novelty and uniqueness of the generated linkers without significant degradation of the most important metrics. Examples of linkers generated by DiffLinker for different input fragments are provided in Figure 6.

**Multiple Fragments**  The major advantage of DiffLinker compared to recently proposed autoregressive models DeLinker and 3DLinker is one-shot generation of the whole linker between arbitrary amounts of fragments. This overcomes the limitation of DeLinker and 3DLinker, which can only link two fragments at a time. Although these autoregressive models can be adjusted to connect pairs of fragments iteratively while growing the molecule, the full context cannot be taken into account in this case. Therefore, suboptimal solutions are more likely to be produced. To illustrate this difference, we adapted 3DLinker to iteratively connect pairs of fragments in molecules where more than two fragments should be connected. As shown in Table 1, 3DLinker fails to construct valid molecules in almost 84% of cases and cannot recover any reference molecule while, despite the higher complexity of linkers in this dataset, our models achieve 93% validity and recover more than 85% of the reference molecules. Besides, molecules generated by 3DLinker have no rings in the linkers, have substantially lower QED and are much harder to synthesize. Examples of linkers generated by DiffLinker for different input fragments are provided in Figure 5. An example of the DiffLinker sampling process for the molecule from the GEOM dataset is shown in Figure 1.

**Protein Pocket Conditioning**  To illustrate the ability of DiffLinker to take surrounding pockets into account, we trained three models on the Pockets Dataset: these are respectively conditioned on the full-atomic pocket representation, conditioned on the pocket backbone atoms and unconditioned. Besides the standard metrics reported in Tables 8 and 9, we also compute the number of clashes between generated molecules and surrounding pockets. We say that there is a clash between two atoms if the distance between them is lower than the sum of their van der Waals radii. As shown in Figure 2, the model conditioned on the full-atomic pocket representation generates molecules with almost the same amount of clashes (in average 7 clashes per molecule) as in the reference complexes from the test set (in average 6 clashes per molecule). There is a clear trend on the number of clashes depending on the amount of information about pockets DiffLinker is conditioned on: the model conditioned on pocket backbone atoms generates molecules with 14 clashes in average, and the unconditioned model produces molecules with 21 clashes in average.

## 6 CONCLUSION

In this work, we introduced DiffLinker, a new E(3)-equivariant 3D-conditional diffusion model for molecular linker design. DiffLinker designs realistic molecules from a set of disconnected fragments by generating a linker, i.e., an atomic point cloud that interconnects the input fragments. *While previous methods were only capable to connect pairs of fragments, DiffLinker naturally scales to an arbitrary number of fragments.* Our method does not require to specify the attachment points of the fragments and predicts the distribution of linker size from the fragments. We show that the proposed method outperforms other models on standard benchmarks and produces more chemically-relevant molecules. *Besides, we demonstrate that our model can be conditioned on protein pockets and generate linkers with a minimum number of clashes.* We believe that our method will accelerate the development of prospective drug candidates and has the potential to become widely used in the fragment-based drug design community.

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

# A APPENDIX

---

**Algorithm 1** Training

---

**Input:** linker $x$, context $u$, neural network $\varphi$
Sample $t \sim \mathcal{U}(0, \ldots, T)$, $\epsilon_t \sim \mathcal{N}(0, \mathbf{I})$
$z_t \leftarrow \alpha_t x + \sigma_t \epsilon_t$
$\hat{\epsilon}_t \leftarrow \varphi(z_t, u, t)$
Minimize $\|\epsilon - \hat{\epsilon}_t\|^2$

---

---

**Algorithm 2** Sampling

---

**Input:** context $u$, neural network $\varphi$
Center context $u$ at $f(u)$
Sample $z_T \sim \mathcal{N}(0, \boldsymbol{I})$
**for** $t$ in $T$, $T-1$, $\ldots$ 1:
    Sample $\epsilon_t \sim \mathcal{N}(0, \boldsymbol{I})$
    $\hat{\epsilon}_t \leftarrow \varphi(z_t, u, t)$
    $z_{t-1} \leftarrow (1/\overline{\alpha}_t) \cdot z_t - \overline{\sigma}_t^2/(\overline{\alpha}_t \sigma_t) \cdot \hat{\epsilon}_t + \varsigma_t \cdot \epsilon$
**end for**
Sample $x \sim p(x|z_0, u)$

---

## A.1 APPLICATIONS

There are several major drug discovery directions in which one of highly relevant or even preferred strategies is to design a linker for fragments that are placed in the space and have fixed positions.

**Fragment Based Drug Discovery (FBDD)** By analogy with classical drug discovery methods, one of the common strategies in FBDD is to operate on fragments that strongly interact with the target proteins. First, strongly binding fragments are defined and characterized (using high-throughput screening followed by X-ray / NMR, or virtual screening and docking). As a result, the exact location and orientation of the fragments in which they bind strongly to the parts of the protein pocket is defined. The next step is to find a linker between the fragments that preserves positions and thus the binding strength of the fragments (preferably, adding the linker will boost the binding strength of the whole molecule) Bancet et al. (2020). There has been reported a range of successful works in which the starting point for a linker design was a crystal structure of a protein with fragments bound to it (Bancet et al., 2020). To name a few, inhibitors for CK2 (De Fusco et al., 2017), LDH-A (Kohlmann et al., 2013) and Dot1L (Mobitz et al., 2017), which are proteins playing crucial roles in progress of various cancers, were designed by linking the fragments that were experimentally observed in a bound state with the corresponding targets.

**Proteolysis targeting chimera (PROTAC)** PROTAC is a heterobifunctional small molecule designed for stimulating degradation of a target protein by connecting it to an E3-ligase. PROTAC consists of two ligands joined by a linker: one ligand recruits and binds a target protein while the other recruits and binds E3 ubiquitin ligase (Békés et al., 2022). For designing PROTACs, one of possible strategies is to dock two proteins (with ligands bound to them) to explore a favorable conformation of the prospective tertiary complex. This information about the initial docking pose of the proteins and exact positions of bound fragments is further used for designing a linker that will stabilize the whole complex (Bai et al., 2021; Farnaby et al., 2019).

**Scaffold hopping** Scaffold hopping is a strategy for designing novel compounds by replacing the central core structure of the known molecule. As shown by Sun et al. (2012), various scaffold-hopping strategies rely on the experimental 3D data of the initial compound bound to a target complex: the information about the geometry of the initial bound molecule is important for altering its core with the increase of the binding affinity, potency or selectivity of the whole molecule. In such a case, scaffold-hopping of the bound molecule can be considered as a linking problem of several disconnected fragments with fixed known 3D coordinates.

A.2   PROOF OF PROPOSITION 1

O(3)-equivariance of function $f$ and the fact that $q$ is isotropic Gaussian distribution implies O(3)-equivariance of the prior distribution:

$$p(\boldsymbol{R}\boldsymbol{z}_T|\boldsymbol{R}\boldsymbol{u}) = \mathcal{N}(\boldsymbol{R}\boldsymbol{z}_T|f(\boldsymbol{R}\boldsymbol{u}), \boldsymbol{I}) = \mathcal{N}(\boldsymbol{R}\boldsymbol{z}_T|\boldsymbol{R}f(\boldsymbol{u})) = \mathcal{N}(\boldsymbol{z}_T|f(\boldsymbol{u})) = p(\boldsymbol{z}_T|\boldsymbol{u}).$$

Likewise, O(3)-equivariance of function $\varphi$ and Equation (8) imply O(3)-equivariance of all transition probabilities $p(\boldsymbol{z}_{t-1}|\boldsymbol{z}_t, \boldsymbol{u})$.

To obtain the distribution $p(\boldsymbol{z}_0|\boldsymbol{u})$ of data point $\boldsymbol{z}_0$, we can consider joint distribution $p(\boldsymbol{z}_0, \boldsymbol{z}_1, \dots, \boldsymbol{z}_T|\boldsymbol{u})$ and marginalize it by $\boldsymbol{z}_{1\dots T}$:

$$p(\boldsymbol{z}_0|\boldsymbol{u}) = \int p(\boldsymbol{z}_0, \boldsymbol{z}_1, \dots, \boldsymbol{z}_T|\boldsymbol{u})d\boldsymbol{z}_{1\dots T} = \int p(\boldsymbol{z}_T|\boldsymbol{u}) \prod_{t=0}^{T-1} p(\boldsymbol{z}_t|\boldsymbol{z}_{t+1}, \boldsymbol{u})d\boldsymbol{z}_{1\dots T}.$$

Having prior and all transition distributions equivariant, it is now trivial to show O(3)-equivariance of $p(\boldsymbol{z}_0|\boldsymbol{u})$:

$$\begin{aligned}
p(\boldsymbol{R}\boldsymbol{z}_0|\boldsymbol{R}\boldsymbol{u}) &= \int p(\boldsymbol{R}\boldsymbol{z}_T|\boldsymbol{R}\boldsymbol{u}) \prod_{t=0}^{T-1} p(\boldsymbol{R}\boldsymbol{z}_t|\boldsymbol{R}\boldsymbol{z}_{t+1}, \boldsymbol{R}\boldsymbol{u})d\boldsymbol{z}_{1\dots T} \\
&= \int p(\boldsymbol{z}_T|\boldsymbol{u}) \prod_{t=0}^{T-1} p(\boldsymbol{R}\boldsymbol{z}_t|\boldsymbol{R}\boldsymbol{z}_{t+1}, \boldsymbol{R}\boldsymbol{u})d\boldsymbol{z}_{1\dots T} \quad \text{(equivariant prior } p(\boldsymbol{z}_T|\boldsymbol{u})\text{)} \\
&= \int p(\boldsymbol{z}_T|\boldsymbol{u}) \prod_{t=0}^{T-1} p(\boldsymbol{z}_t|\boldsymbol{z}_{t+1}, \boldsymbol{u})d\boldsymbol{z}_{1\dots T} \quad \text{(equivariant transition kernels } p(\boldsymbol{z}_t|\boldsymbol{z}_{t+1}, \boldsymbol{u})\text{)} \\
&= \int p(\boldsymbol{z}_0, \boldsymbol{z}_1, \dots, \boldsymbol{z}_T|\boldsymbol{u})d\boldsymbol{z}_{1\dots T} = p(\boldsymbol{z}_0|\boldsymbol{u}).
\end{aligned}$$

A.3   PROBLEM WITH TRANSLATIONS

Consider transition probability $p(\boldsymbol{z}_{t-1}|\boldsymbol{z}_t, \boldsymbol{u}) = q(\boldsymbol{z}_{t-1}|\hat{\boldsymbol{x}}, \boldsymbol{z}_t)$. Translation equivariance of $p(\boldsymbol{z}_{t-1}|\boldsymbol{z}_t, \boldsymbol{u})$ means that

$$p(\boldsymbol{z}_{t-1} + \boldsymbol{t}|\boldsymbol{z}_t + \boldsymbol{t}, \boldsymbol{u} + \boldsymbol{t}) = p(\boldsymbol{z}_{t-1}|\boldsymbol{z}_t, \boldsymbol{u}) \ \forall \boldsymbol{t} \in \mathbb{R}^3. \tag{13}$$

More precisely,

$$\mathcal{N}(\boldsymbol{z}_{t-1} + \boldsymbol{t}; \hat{\boldsymbol{\mu}}_t(\boldsymbol{z_t} + \boldsymbol{t}, \boldsymbol{u} + \boldsymbol{t}), \varsigma_t^2 \boldsymbol{I}) = \mathcal{N}(\boldsymbol{z}_{t-1}; \hat{\boldsymbol{\mu}}_t(\boldsymbol{z_t}, \boldsymbol{u}), \varsigma_t^2 \boldsymbol{I}), \tag{14}$$

where

$$\hat{\boldsymbol{\mu}}_t(\boldsymbol{z_t}, \boldsymbol{u}) = \boldsymbol{\mu}_t(\hat{\boldsymbol{x}}, \boldsymbol{z_t}) = \frac{\overline{\alpha}_t \sigma_{t-1}^2}{\sigma_t^2} \boldsymbol{z}_t + \frac{\alpha_{t-1}\overline{\sigma}_t^2}{\sigma_t^2} \hat{\boldsymbol{x}}, \tag{15}$$

and

$$\hat{\boldsymbol{x}} = \frac{1}{\alpha_t} \boldsymbol{z}_t - \frac{\sigma_t}{\alpha_t} \varphi(\boldsymbol{z_t}, \boldsymbol{u}, t). \tag{16}$$

Therefore, the mean of this distribution can be written as:

$$\hat{\boldsymbol{\mu}}_t(\boldsymbol{z_t}, \boldsymbol{u}) = \frac{1}{\overline{\alpha}_t} \boldsymbol{z}_t - \frac{\overline{\sigma}_t^2}{\overline{\alpha}_t \sigma_t} \varphi(\boldsymbol{z_t}, \boldsymbol{u}, t). \tag{17}$$

Neural network $\varphi$ is translation invariant meaning that $\varphi(\boldsymbol{z}_t + \boldsymbol{t}, \boldsymbol{u} + \boldsymbol{t}, t) = \varphi(\boldsymbol{z}_t, \boldsymbol{u}, t)$.

It means that:

$$\hat{\boldsymbol{\mu}}_t(\boldsymbol{z_t} + \boldsymbol{t}, \boldsymbol{u} + \boldsymbol{t}) = \frac{1}{\overline{\alpha}_t}(\boldsymbol{z_t} + \boldsymbol{t}) - \frac{\overline{\sigma}_t^2}{\overline{\alpha}_t \sigma_t} \varphi(\boldsymbol{z_t} + \boldsymbol{t}, \boldsymbol{u} + \boldsymbol{t}, t) \tag{18}$$

$$= \frac{1}{\overline{\alpha}_t}(\boldsymbol{z_t} + \boldsymbol{t}) - \frac{\overline{\sigma}_t^2}{\overline{\alpha}_t \sigma_t} \varphi(\boldsymbol{z_t}, \boldsymbol{u}, t) \tag{19}$$

$$= \frac{1}{\overline{\alpha}_t}\boldsymbol{z_t} - \frac{\overline{\sigma}_t^2}{\overline{\alpha}_t \sigma_t} \varphi(\boldsymbol{z_t}, \boldsymbol{u}, t) + \frac{1}{\overline{\alpha}_t}\boldsymbol{t} \tag{20}$$

$$= \hat{\boldsymbol{\mu}}_t(\boldsymbol{z_t}, \boldsymbol{u}) + \frac{1}{\overline{\alpha}_t}\boldsymbol{t} \tag{21}$$

$$= \hat{\boldsymbol{\mu}}_t(\boldsymbol{z_t}, \boldsymbol{u}) + \lambda_t \boldsymbol{t}. \tag{22}$$

So we see that $\hat{\boldsymbol{\mu}}_t(\boldsymbol{z_t}, \boldsymbol{u})$ is equivariant to translations, however input and output translations **are not equal** because $\lambda \neq 1$. It means that equivariance of distributions from Equations (13) and (14) does not hold. More formally,

$$p(\boldsymbol{z}_{t-1} + \boldsymbol{t} | \boldsymbol{z}_t + \boldsymbol{t}, \boldsymbol{u} + \boldsymbol{t}) = \mathcal{N}\left(\boldsymbol{z}_{t-1} + \boldsymbol{t}; \hat{\boldsymbol{\mu}}_t(\boldsymbol{z_t}, \boldsymbol{u}) + \lambda \boldsymbol{t}, \varsigma_t^2 \boldsymbol{I}\right) \tag{23}$$

$$= \mathcal{N}\left(\boldsymbol{z}_{t-1}; \hat{\boldsymbol{\mu}}_t(\boldsymbol{z_t}, \boldsymbol{u}) + (\lambda - 1)\boldsymbol{t}, \varsigma_t^2 \boldsymbol{I}\right) \tag{24}$$

$$\neq \mathcal{N}\left(\boldsymbol{z}_{t-1}; \hat{\boldsymbol{\mu}}_t(\boldsymbol{z_t}, \boldsymbol{u}), \varsigma_t^2 \boldsymbol{I}\right). \tag{25}$$

We can also write that

$$p(\boldsymbol{z}_{t-1} + \boldsymbol{t} | \boldsymbol{z}_t + \boldsymbol{t}, \boldsymbol{u} + \boldsymbol{t}) = p\left(\boldsymbol{z}_{t-1} + (1 - \lambda)\boldsymbol{t} | \boldsymbol{z}_t, \boldsymbol{u}\right) \neq p\left(\boldsymbol{z}_{t-1} | \boldsymbol{z}_t, \boldsymbol{u}\right). \tag{26}$$

### A.4 DIFFUSION MODEL

We trained all DiffLinker[1] models with $T = 500$ diffusion steps using polynomial noise schedule:

$$\alpha_t = (1 - 2s) \cdot (1 - (t/T)^2), \tag{27}$$

where $s = 10^{-5}$ is a precision value that helps to avoid numerically unstable situations (Hoogeboom et al., 2022).

**Sampling**  For all the experiments discussed in the main text, we sampled with the same number of denoising steps $T = 500$ as used in training. Sampling time for all the datasets is provided in Table 3. Although the time reported in Table 3 is more than affordable for applying our method in practice, we explored the capability of DiffLinker to sample even faster without performance loss. Following Nichol & Dhariwal (2021), we conducted additional evaluation of DiffLinker with the reduced number of denoising steps $T = 500$ in sampling, considering $T/2, T/5, T/10, T/20, T/50$ and $T/100$ values. Figure 3 shows how performance metrics obtained on ZINC test set depend on the number of denoising steps performed in sampling. In all cases we used DiffLinker pretrained on ZINC with $T = 500$ denoising steps. As shown in Figure 3, our model is robust to significant reduction of the number of denoising steps in sampling resulting in 10-fold gain in sampling speed without any performance degradation. Effectively, one can reduce the sampling speed from 0.365 to 0.036 seconds per molecule with no significant performance metrics loss.

### A.5 DYNAMICS

EGNN takes as input a graph of atoms belonging to the linker $\boldsymbol{z}_t$ and its context $\boldsymbol{u}$ represented by feature vectors $\boldsymbol{h}_i \in \mathbb{R}^{\text{in}}$ and coordinates $\boldsymbol{r}_i \in \mathbb{R}^3$. Feature vector $\boldsymbol{h}_i$ consists of atom types, fragments flag and time step $t$. If anchors are known, additionally anchor flag is passed. If the model is conditioned on the protein pocket, additionally pocket flag is passed.

First, atom features are passed to the encoder: $\boldsymbol{h}_i \rightarrow \text{Linear(in, nf)} \rightarrow \boldsymbol{h}_i^0$.

Next, as discussed in Section 4.2, $L$ Equivariant Graph Convolutional Layers (EGCL) are sequentially applied. Learnable components of EGCL $\phi_e, \phi_h, \phi_r$ are implemented as neural networks that include fully-connected layers (FC), batch normalization layers (BN) and activations SiLU.

---

[1]DiffLinker code is available at: https://anonymous.4open.science/r/DiffLinker

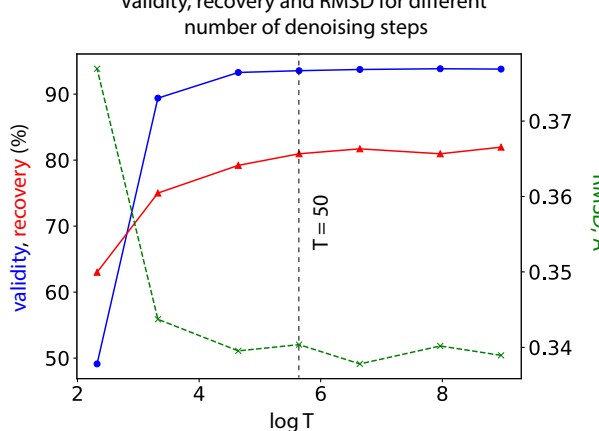

Figure 3: Dependency of validity, recovery and RMSD on the number of denoising steps in sampling shows that DiffLinker is robust to reducing the number of denoising steps. The robustness of DiffLinker allows for 10-fold gain in sampling speed without any performance degradation. For all experiments we used DiffLinker trained on ZINC with $T = 500$ steps and performed evaluation on ZINC test set sampling 250 linkers for each input set of fragments.

Table 3: Sampling time for different datasets (with $T = 500$ denoising steps). Experiments were performed on a single Tesla V100-PCIE-32GB GPU.

| dataset | batch size | time per molecule, s |
|---|---|---|
| ZINC | 128 | 0.37 |
| CASF | 128 | 0.63 |
| Multi-Fragment | 64 | 0.37 |
| Pockets (full atomic representation) | 32 | 0.46 |
| Pockets (backbone representation) | 32 | 0.21 |
| Pockets (no pocket) | 32 | 0.12 |

**Message** $\phi_e$: takes a pair of node embeddings $\boldsymbol{h}_i^l$ and $\boldsymbol{h}_j^l$ and the squared distance $d_{ij}^2 = \|\boldsymbol{r}_i - \boldsymbol{r}_j\|^2$ between these nodes and outputs a message $\boldsymbol{m}_{ij} \in \mathbb{R}^{\text{nf}}$:

$$\text{concat}[\boldsymbol{h}_i^l, \boldsymbol{h}_j^l, d_{ij}^2] \rightarrow \{\text{FC}(2 \cdot \text{nf} + 1, \text{nf}) \rightarrow \text{SiLU} \rightarrow \text{FC}(\text{nf}, \text{nf}) \rightarrow \text{SiLU}\} \rightarrow \boldsymbol{m}_{ij}$$

**Features update** $\phi_h$: takes as input node embedding $\boldsymbol{h}_i^l$ and its aggregated message $\boldsymbol{m}_i = \sum_j \boldsymbol{m}_{ij}$ and returns the updated node embedding:

$$\text{concat}[\boldsymbol{h}_i^l, \boldsymbol{m}_i] \rightarrow \{\text{FC}(2 \cdot \text{nf}, \text{nf}) \rightarrow \text{BN} \rightarrow \text{SiLU} \rightarrow \text{FC}(\text{nf}, \text{nf}) \rightarrow \text{BN} \rightarrow \text{add}(\boldsymbol{h}_i^l)\} \rightarrow \boldsymbol{h}_i^{l+1}$$

**Coordinates update** $\phi_r$: takes the same input as $\phi_e$ and outputs a scalar value

$$\text{concat}[\boldsymbol{h}_i^l, \boldsymbol{h}_j^l, d_{ij}^2] \rightarrow \{\text{FC}(2 \cdot \text{nf} + 1, \text{nf}) \rightarrow \text{SiLU} \rightarrow \text{FC}(\text{nf}, \text{nf}) \rightarrow \text{SiLU} \rightarrow \text{FC}(\text{nf}, 1)\} \rightarrow \text{output}$$

**Training** We trained separate models for ZINC, Multi-Frag and Pocket datasets. Hyper parameters of the models and average time required for training one epoch are provided in Table 4. All models were trained on a single Tesla V100-PCIE-32GB GPU using Adam with learning rate $2 \cdot 10^{-5}$ and weight decay $10^{-13}$.

### A.6 LINKER SIZE PREDICTION

Graph neural network for predicting probabilities of the number of atoms in the prospective linker for a given set of fragments takes as input a fully-connected graph of atoms belonging to the fragments represented by feature vectors $\boldsymbol{h}_i \in \mathbb{R}^{\text{in}}$ and inter-atomic squared distances $d_{ij}^2 = \|\boldsymbol{r}_i - \boldsymbol{r}_j\|^2$, and outputs a vector of probabilities corresponding to the predefined linker sizes $\boldsymbol{p} \in [0, 1]^{\text{out}}$.

First, node embeddings are computed: $\boldsymbol{h}_i \rightarrow \text{Linear}(\text{in}, \text{nf}) \rightarrow \boldsymbol{h}_i^0$.

Table 4: Hyper parameters of EGNN models trained on ZINC, Multi-Fragment and Pocket datasets.

| dataset | given anchors | nf | $L$ | batch size | epochs | time per epoch, min |
|---|---|---|---|---|---|---|
| ZINC | no | 128 | 8 | 128 | 300 | 15.2 |
| ZINC | yes | 128 | 8 | 128 | 300 | 17.1 |
| Multi-Fragment | no | 128 | 6 | 64 | 839 | 10.6 |
| Multi-Fragment | yes | 128 | 6 | 128 | 1240 | 10.1 |
| Pocket (full atomic representation) | yes | 128 | 6 | 32 | 420 | 20.3 |
| Pocket (backbone representation) | yes | 128 | 6 | 32 | 620 | 10.7 |
| Pocket (no pocket) | yes | 128 | 6 | 32 | 670 | 8.4 |

Table 5: Hyper parameters of SizeGNN models trained on ZINC and Multi-Fragment datasets.

| dataset | in | hid | out | $L$ | batch size | epochs | time per epoch, min |
|---|---|---|---|---|---|---|---|
| ZINC | 8 | 256 | 10 | 5 | 256 | 53 | 10.6 |
| Multi-Fragment | 9 | 256 | 33 | 5 | 256 | 119 | 9.2 |

Next, a sequence of $L$ Graph Convolutional Layers (GCL) is applied. Learnable components of GCL $\phi_e$, $\phi_h$ are implemented in the same way as for EGNN.

Finally, node embeddings $h_i^L$ are projected onto $\mathbb{R}^{\text{out}}$, aggregated and normalized resulting in the vector of label probabilities:

$$h_i^L \rightarrow \{\text{FC(nf, out)} \rightarrow \text{Mean} \rightarrow \text{Softmax}\} \rightarrow p$$

**Training** We trained two models for ZINC and Multi-Frag datasets. Hyper parameters of the models and average time required for training one epoch are provided in Table 5. Both models were trained using Adam with learning rate $10^{-4}$ and weight decay $10^{-13}$. Both models were trained on a single Tesla V100-PCIE-32GB GPU.

## A.7 3DLINKER ON GEOM DATASET

In order to run 3DLinker on GEOM dataset, we had to additionally filter the original test set consisting of 1,288 input fragment sets and remove examples with more than 3 disconnected fragments. For the remainder test set that included 1,170 input fragment triplets, we ran 3DLinker twice: first, to connect two randomly selected fragments (10 samples per fragment pair) and then to connect the resulting compound with the third fragment (10 samples per input). In both steps, we used the half of the original linker size. Overall, we obtained 100 samples for each input fragment triplet. We used a pre-trained 3DLinker model available at `https://github.com/YinanHuang/3DLinker`. The results are provided in Tables 1 and 2.

## A.8 EVALUATION DETAILS

The principal difference between our and other methods is that we generate 3D point cloud of atoms that should be further connected with covalent bonds while other methods generate covalent bonds along with atom types. We emphasize that both DeLinker and 3DLinker employ valency rules at each generation step which makes is much easier to achieve high validity of samples. In our case, DiffLinker learns these chemical rules from the data and places atoms at the relevant distances from each other. Since the output of DiffLinker is a 3D point cloud, we need to additionally compute covalent bonds between pairs of atoms based on their types and pairwise distances. For doing that, we use OpenBabel (O'Boyle et al., 2011).

To be consistent in the evaluation methodology, we recomputed covalent bonds using OpenBabel for all molecules in ZINC and CASF test sets. Next, for each updated molecule, we obtained linkers by removing irrelevant atoms and saved the resulting molecules and fragments in SDF and SMILES formats. Molecules saved in SDF format were considered as ground truth and used for 3D comparison

Table 6: QED, SA and number of rings for molecules in training, validation and test datasets.

|  | Dataset | QED ↑ | SA ↓ | # Rings ↑ |
|---|---|---|---|---|
| ZINC | Train | 0.734 | 2.986 | 0.213 |
| | Validation | 0.736 | 2.930 | 0.207 |
| | Test | 0.729 | 2.944 | 0.274 |
| GEOM | Train | 0.546 | 2.720 | 0.901 |
| | Validation | 0.556 | 2.711 | 0.853 |
| | Test | 0.537 | 2.735 | 0.849 |
| Pockets | Train | 0.330 | 5.330 | 0.926 |
| | Validation | 0.418 | 4.783 | 0.763 |
| | Test | 0.356 | 5.445 | 1.272 |
| | CASF | 0.481 | 3.572 | 0.271 |

(for computing RMSD and $SC_{RDKit}$ metrics). Molecules and linkers saved in SMILES format were considered as ground truth and used for 2D comparison (novelty and recovery rates). To evaluate other methods, we used original SMILES representations.

**Our samples**  For each generated point cloud, we computed covalent bonds with OpenBabel, and extracted the largest connected component. Next, we obtained a linker by matching the generated molecule with the corresponding fragments (computed with OpenBabel as explained above) and removing irrelevant atoms. Finally, we kekulized the resulting linker and saved the generated molecule with recomputed covalent bonds and the corresponding linker in SDF and SMILES formats.

**Metrics**  To compute validity, we apply sanitization and additionally check that the molecule contains all atoms from fragments. For all other metrics, we consider only a subset of valid samples. To compute novelty, we first preprocess SMILES of the linker by removing stereochemistry and canonicalizing tautomer SMILES. Then, we count how many of the resulting generated linker SMILES were represented in the training set. To compute uniqueness, we compare SMILES of whole molecules and count number of unique molecules sampled for each input pair of fragments. To compute recovery, we compare SMILES of each molecule sampled for a given pair of fragments with SMILES of the corresponding ground-truth molecule. Before comparison, we remove hydrogens and stereochemistry from molecules. To compute RMSD, we consider only recovered molecules and align them with the corresponding ground-truth molecules using RDKit function `rdkit.Chem.rdMolAlign` which returns the optimal RMSD for aligning two molecules. To compute quantitative drug-likeness (QED) (Bickerton et al., 2012), we used RDKit function `rdkit.Chem.QED.qed`. To compute synthetic accessibility, we used function `calculateScore` provided by Ertl & Schuffenhauer (2009) in the RDKit-compatible package `sascorer.py`. For calculating the number of rings, we used RDKit function `rdkit.Chem.rdMolDescriptors.CalcNumRings`. Table 6 provides mean QED, SA and number of rings computed for molecules in training, validation and test datasets.

### A.9  Additional Results

As explained in Section A.7, we had to additionally filter the original GEOM test set consisting of 1,288 input fragment sets and remove examples with more than 3 disconnected fragments. For each input fragment set, we obtained 100 samples with 3DLinker. For consistency we report DiffLinker performance on GEOM in Tables 1 and 2 computed in the same setting: 1,170 input examples and 100 samples per input. DiffLinker results computed on the full GEOM test set with 250 samples per input are provided in Tables 8 and 9.

DiffLinker results on the Pockets test set with 100 samples per input are provided in Tables 8 and 9 as well. We note that train/test split of the Pockets dataset was performed solely based on PDB-codes of the protein-ligand complexes and EC numbers of proteins. In the resulting dataset used for evaluation, we therefore have 17 molecules that are also represented in the training set but bound to

different proteins. For the full picture, we alternatively provide another reduced test set which does not contain molecules from the training set at all. It includes 453 examples from the initial test set. We provide evaluation metrics obtained on the reduced test set in Table 10.

## A.10 INPAINTING

Table 7: Comparison of DiffLinker trained in 3D-conditioning setting with DiffLinker trained in inpainting setting. Evaluation was performed on ZINC validation set. For each input pair of fragments we sampled 50 linkers.

| Method | Valid, % | Recovered, % | RMSD ↓ | SC$_{RDKit}$ ↑ | QED ↑ | SA ↓ | # Rings ↑ |
|---|---|---|---|---|---|---|---|
| 3D-conditioning | **91.0** | **66.3** | **0.31** | **0.92** | 0.71 | **3.02** | 0.21 |
| Inpainting | 65.5 | 65.9 | 0.79 | 0.89 | **0.73** | 3.09 | **0.25** |

An alternative way of conditioned linker generation with diffusion models is *inpainting* strategy (Lugmayr et al., 2022), in which the model is trained to denoise a full molecule, while at the inference step the known part of the molecule is generated using the true denoising process. Inpainting strategy for molecular linker design can be easily implemented using vanilla EDM (Hoogeboom et al., 2022) with a slight tweak of the sampling function.

This approach was the first that we tried in our experiments. However, models trained in 3D-conditioning setting, explained in Section 4.1, significantly outperformed the inpainting models. In Table 7 we provide a comparison of DiffLinker trained in 3D-conditioning setting with DiffLinker trained in inpainting setting. Both models have identical architectures and were trained with identical hyper parameters. For evaluation we used validation ZINC set (400 examples) and sampled 50 linkers for every pair of input fragments.

As shown in Table 7, using 3D-conditioned model reduces the number of invalid molecules (i.e., chemically incorrect or with disconnected fragments) by 25%. Besides, inpainting model generates molecules with more than 2-fold higher RMSD. Even though the chemical properties of the molecules generated by both methods are comparable, significantly lower validity indicates that inpainting approach is suboptimal in the molecular linker design task.

## A.11 2D FILTERS

2D Filters used by Imrie et al. (2020) for constructing ZINC and CASF datasets include synthetic accessibility (Ertl & Schuffenhauer, 2009), ring aromaticity (RA), and pan-assay interference compounds (PAINS) (Baell & Holloway, 2010) criteria. RA controls the correctness of the covalent bond orders in the rings of the a linker and PAINS checks if a molecule does not contain compounds that often give false-positive results in high-throughput screens (Baell & Holloway, 2010). Even though we used the same datasets as in Imrie et al. (2020) that were created using all three filters, we however modify the metric "Passed 2D Filters" by removing SA from it. Instead we introduce our own SA-based metric that we report separately.

**Problem with SA filter used by Imrie et al. (2020) and Huang et al. (2022)** The molecule is considered to pass the synthetic accessibility filter if its SA-score is lower than SA-score of the corresponding pair of fragments. Even though our models performed on par or better than DeLinker and 3DLinker according to all other metrics, almost all molecules generated by our models did not pass SA-filter. We investigated this issue and figured out that SMILES of fragments passed to SA filter by DeLinker and 3DLinker contained dummy atoms representing anchors. These atoms did not have any atom type assigned and therefore such molecules were considered hard to be synthesised. For almost all molecules in the test set SA-scores of fragments with dummy atoms were higher than SA-scores of the whole molecules. However, for most of fragments without dummy atoms SA-scores were much lower. Figure 4 shows 4 examples of molecules and fragments with and without dummy atoms and the corresponding SA-scores. We can conclude that SA-filter proposed by authors of DeLinker shows nothing but the fact that molecules with unknown atoms are hard to be synthesized. Therefore, we considered this metric to be irrelevant and excluded it from our report. Instead, we report average Synthetic Accessibility score of full generated molecules.

Table 8: Performance metrics on GEOM (250 samples, full test set) and Pockets test set (100 samples). The first three metrics assess the chemical relevance of the generated molecules. The last three metrics evaluate the standard generative properties of the methods.

| | Method | QED ↑ | SA ↓ | # Rings ↑ | Valid, % | Unique, % | Novel, % |
|---|---|---|---|---|---|---|---|
| GEOM | DiffLinker | **0.48** | **2.99** | 0.75 | **93.4** | 31.6 | 68.7 |
| | DiffLinker (given anchors) | **0.49** | **3.01** | **0.79** | **93.5** | 32.1 | 68.5 |
| | DiffLinker (sampled size) | 0.45 | 3.27 | 0.76 | 87.1 | **57.3** | **76.2** |
| | DiffLinker (given anchors, sampled size) | 0.46 | 3.33 | **0.84** | 88.6 | **58.2** | **76.2** |
| Pockets | DiffLinker (pocket atoms) | 0.45 | 3.89 | **1.06** | 88.7 | **62.5** | **73.1** |
| | DiffLinker (pocket backbone) | **0.45** | **3.77** | 0.95 | **90.4** | 60.9 | 72.8 |
| | DiffLinker (unconditioned) | **0.45** | 3.83 | **1.10** | **93.6** | 61.1 | **74.9** |

Table 9: Metrics assessing the ability of the methods to generate molecules that are chemically and geometrically similar to the reference ones. For GEOM (250 samples, full test set) and Pockets test set (100 samples).

| | | | | | SC$_{RDKit}$ | | | |
|---|---|---|---|---|---|---|---|---|
| | Method | 2D Filters, % | Recovery, % | RMSD ↓ | > 0.7 | > 0.8 | > 0.9 | Avg ↑ |
| GEOM | DiffLinker | 49.47 | **89.1** | 0.11 | **95.90** | **88.81** | **73.31** | **0.93** |
| | DiffLinker (given anchors) | 51.25 | **88.0** | 0.11 | **95.57** | **87.77** | **72.12** | **0.93** |
| | DiffLinker (sampled size) | **56.05** | 77.5 | **0.07** | 92.11 | 82.11 | 58.03 | 0.88 |
| | DiffLinker (given anchors, sampled size) | **58.16** | 77.1 | **0.07** | 91.38 | 80.37 | 55.71 | 0.88 |
| Pockets | DiffLinker (pocket atoms) | **69.47** | 37.3 | **0.87** | **71.61** | **57.30** | **34.52** | **0.78** |
| | DiffLinker (pocket backbone) | **66.48** | 37.8 | 0.89 | **65.58** | **52.77** | **31.50** | **0.76** |
| | DiffLinker (unconditioned) | 59.43 | 36.5 | **0.89** | 64.06 | 50.95 | 30.59 | 0.75 |

Table 10: Performance metrics on the reduced Pockets test set (453 examples, 100 samples).

| Method | QED ↑ | SA ↓ | # Rings ↑ | Valid, % | Unique, % | Novel, % |
|---|---|---|---|---|---|---|
| DiffLinker (pocket atoms) | 0.50 | 3.77 | **1.07** | 86.2 | **77.5** | **90.7** |
| DiffLinker (pocket backbone) | 0.50 | 3.62 | 0.94 | **88.3** | 75.6 | 90.6 |
| DiffLinker (unconditioned) | 0.51 | 3.70 | **1.13** | **92.0** | 75.7 | **93.0** |

| | | | | SC$_{RDKit}$ | | | |
|---|---|---|---|---|---|---|---|
| Method | 2D Filters, % | Recovery, % | RMSD ↓ | > 0.7 | > 0.8 | > 0.9 | Avg ↑ |
| DiffLinker (pocket atoms) | **60.84** | **31.2** | **0.86** | **66.35** | **48.67** | **26.62** | **0.75** |
| DiffLinker (pocket backbone) | **57.15** | 31.1 | 1.03 | **58.76** | **42.81** | **23.14** | **0.72** |
| DiffLinker (unconditioned) | 48.52 | 30.1 | **0.96** | 57.03 | 40.57 | 22.16 | 0.72 |

| | | | | |
|---|---|---|---|---|
| our fragments (without dummy atoms) | 2.11 | 2.01 | 2.11 | 2.33 |
| full molecules | 2.29 | 2.23 | 2.88 | 2.21 |
| DeLinker fragments (with dummy atoms) | 3.26 | 3.49 | 4.37 | 3.44 |

Figure 4: Synthetic accessibility scores (SA-scores) for fragments without dummy atoms (top row), full molecules (middle row) and fragments with dummy atoms (bottom row).

Figure 5: Examples of linkers generated by DiffLinker (sampled size) for fragments from GEOM datasets.

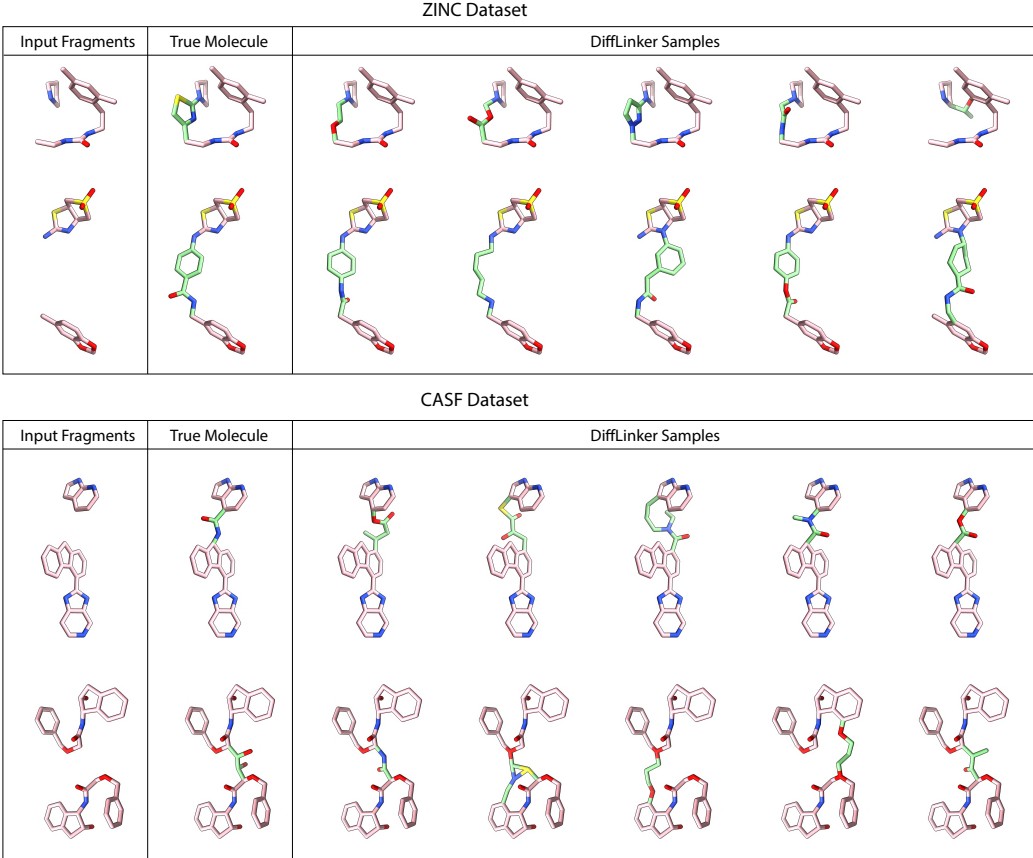

Figure 6: Examples of linkers generated by DiffLinker (sampled size) for fragments from CASF and ZINC datasets.

