# OpenReview forum: "Equivariant 3D-Conditional Diffusion Models for Molecular Linker Design"
_ICLR.cc/2023/Conference — Submitted to ICLR 2023_

### Official Review · Reviewer_oTUZ · 2022-10-20

**Confidence:** 4
**Correctness:** 4
**Technical Novelty And Significance:** 2
**Empirical Novelty And Significance:** 3
**Recommendation:** 6

**Clarity, Quality, Novelty And Reproducibility:**

Code is provided.

The paper is clearly written and related work is well described.

In section 2.1, can the same molecule appear, differently fragmented, in both training and test sets?


**Strength And Weaknesses:**

This is a neat application of equivariant diffusion. The authors have chosen their problem well if the aim is to quickly produce something of practical use for drug discovery.  Linker design is a real task in the drug discovery process, it is constrained enough to do well, and existing ML models for the purpose have limitations that the new model addresses.

Why did the authors choose to train a conditional generator on fragmented molecules, rather than training a model to denoise entire molecules and treating conditional generation as an inpainting task?

The generated molecules in figure 2 look odd e.g. they contain 3, 4, and 7-member rings. Do you get better molecules after some simple post-filtering? Also in this figure, the bonds in the input fragments and true molecule do not look correct.

In table 1, please add the QED, SA and #rings scores for the ZINC, CASF and GEOM training data as well as the generated molecules.


**Summary Of The Paper:**

The authors propose a method to generate the remainder of a molecule in 3D given some fragments. The generator is an E(3) equivariant denoising diffusion model, conditioned on the positions of the fragment atoms, and optionally also on the protein pocket that the molecule should fit into.

**Summary Of The Review:**

The paper combines conditional generation, denoising diffusion, and equivariant GNNs to tackle an important problem in drug discovery.

---

> ### Author Response · Authors · 2022-11-11
> **Response to Reviewer oTUZ**
>
> We thank the reviewer for the positive feedback and we are glad that the reviewer appreciates the relevance of our work for drug discovery. Below we address all the questions and comments raised by the reviewer. Following the reviewer’s recommendations, we introduced all the necessary changes to the submission text.
>
> > Why did the authors choose to train a conditional generator on fragmented molecules, rather than training a model to denoise entire molecules and treating conditional generation as an inpainting task?
>
> In the beginning, we started with the inpainting strategy as it can easily be implemented using a vanilla equivariant diffusion model (EDM) [1] with a slight tweak of the sampling function. However, later we realized that our novel mechanism of 3D-conditioning, where a part of atoms remains unchanged throughout the denoising process, simplifies requirements for E(3)-equivariance (e.g., no need to operate on a center-of-mass-free (CoM-free) system). Besides, 3D-conditioning empirically outperforms the inpainting approach: **it increases the validity of the generated molecules by 25%, and improves RMSD more than two-fold.** We added Section A.10 where we discuss the inpainting strategy and the results in Table 7.
>
> > The generated molecules in figure 2 look odd e.g. they contain 3, 4, and 7-member rings. Do you get better molecules after some simple post-filtering?
>
> Indeed, we might have picked better molecules for this figure – thank you for spotting it. To address your comment, we now performed a ranking by synthetic accessibility and manually selected molecules from the top of the list that would represent existence or lack of clashes with the protein pocket in the most visually clear way.
>
> > Also in this figure, the bonds in the input fragments and true molecule do not look correct.
>
> Thank you for pointing this out. By mistake we used wrong smiles representations for input fragments and the reference molecule for generating these images. Now bonds are correctly assigned.
>
> > In table 1, please add the QED, SA and #rings scores for the ZINC, CASF and GEOM training data as well as the generated molecules.
>
> We computed QED, SA and #rings scores for ZINC (train,test,val), GEOM (train,test,val), Pockets (train,test,val) and CASF (test) and reported them in a separate Table 5. Please let us know if you think it is better to add some of these values specifically in Table 1.
>
> > In section 2.1, can the same molecule appear, differently fragmented, in both training and test sets?
>
> For ZINC data, we do not have any molecules that appear both in train and test sets. None of CASF molecules appears in the ZINC train set as well.
>
> Regarding the GEOM dataset, we detected one molecule (with two different fragmentations) that was represented in both train and test datasets. This was definitely made by mistake, as the code is supposed to split the data by the entire molecule. Now we fixed the test set (i.e., reduced it from 1290 input examples to 1288 examples excluding this molecule with its two fragmentations). We also recomputed all results connected with the GEOM dataset and updated corresponding Tables 1, 2, 8 and 9. We should note there was no significant change in the results.
>
> For the Pocket dataset, we split by protein-ligand complexes, taking into account EC numbers of the proteins, which means that we do not have the same (and similar) proteins in both train and test sets. However, we have 17 small molecules that are represented in both train and test sets, but bind to different proteins. In the case of this dataset, it was expected as we were primarily focused on the diversity of the whole protein-ligand complexes. But for the full picture, we now additionally consider the reduced test set removing these 17 molecules represented in the training set. We perform additional evaluation on the reduced test set and provide results in a new Table 10.
>
> **References:**
>
> [1] Hoogeboom E. et al. Equivariant diffusion for molecule generation in 3D

---

### Official Review · Reviewer_TyBL · 2022-10-22

**Confidence:** 4
**Clarity, Quality, Novelty And Reproducibility:** 1. The paper is pretty clear.
2. Pres…
**Correctness:** 3
**Technical Novelty And Significance:** 2
**Empirical Novelty And Significance:** 2
**Recommendation:** 3

**Strength And Weaknesses:**

[Strength]:

1. The paper is well organized. The tackled linker generation scenario is clearly defined, and the method is easy to follow and understand.
2. The author nicely provides the source code.
3. The benchmark table's numerical results are superior to existing methods.

[Weakness]

1. Overall, the paper is not surprising to me. From the ML perspective, this work is an incremental adoption of the equivariant diffusion model in the context of linker design, without significant technical challenges.
2. Besides, actually diffusion models itself has the capacity for "inpainting" missing parts. As shown in the original diffusion model paper [1], the diffusion model can naturally recover the missing patches of images. This further indicates this "linker design" is a natural and simple usage of existing diffusion models for molecule generation, making the technical contribution limited.
3. I'm also a little doubtful about the experimental setup. From the domain-specific (biological or chemical) perspective, considering we already have several fragments and want to design the linker for them: why for the different disconnected fragments we can know their relative positions and even exact coordinates before generating the linker (as shown in Fig.1). I think for this linker design problem, we should take these fragments' positions as known, which, however, is even treated as rigid bodies in this paper.
4. I'm also concerned about the "reflection" equivariance imposed on the proposed model. According to my knowledge, the energy of reflected biostructures should be different, which means they shouldn't have reflection symmetry. However, this is not the case for the proposed model.
5. [minor point] Considering this is an application paper, actually I'm not sure whether the evaluation itself is sound. For the dataset part, the data is obtained by manually subtracting fragments from molecular structure datasets. This makes me feel like the benchmark itself is human-designed and far away from real-world challenges. I understand this setup may be already adopted in a few existing benchmarks, but this doesn't mean we can just follow them despite the limitation in existing benchmarks. Maybe better and more realistic benchmarks can also be introduced.

[1] Ho, Jonathan, Ajay Jain, and Pieter Abbeel. "Denoising diffusion probabilistic models." Advances in Neural Information Processing Systems 33 (2020): 6840-6851.

**Summary Of The Paper:**

The paper studied the problem of molecular linker generation, which aims to generate the linker given different individual fragments of a desired molecule/drug. Specifically, the work borrows the idea of the recent equivariant diffusion molecule generative model [1] into this specific "molecular missing part" generation problem. Experiments on several human-designed datasets show that the proposed method is competitive or better than existing methods.

[1] Emiel Hoogeboom, Vıctor Garcia Satorras, Clement Vignac, and Max Welling. Equivariant diffusion for molecule generation in 3d. In International Conference on Machine Learning, pp. 8867–8887. PMLR, 2022.

**Summary Of The Review:**

The paper introduced a diffusion model for linker design. the idea is not surprising for the ML community and the technical contribution is a little limited. The empirical improvement over existing methods is large, but the evaluation benchmarks are also not reasonable enough from my view.

---

> ### Author Response · Authors · 2022-11-11
> **Response to Reviewer TyBL [Part 1]**
>
> We thank the Reviewer for the constructive feedback and are glad that the reviewer found our paper “well organized”, problem “clearly defined”, method “easy to follow and understand”, and results “superior to existing methods”. We address all the comments and questions raised by the Reviewer, providing a deeper analysis of possible DiffLinker applications and discussing applicability of the inpainting formulation for the molecular linker design problem (supporting it with quantitative results). Besides, we commented on the questions about reflection equivariance, evaluation methodology, and overall novelty and contribution of our work. We will be glad to provide additional comments in case the Reviewer has any follow-up questions.
>
> > Overall, the paper is not surprising to me. From the ML perspective, this work is an incremental adoption of the equivariant diffusion model in the context of linker design, without significant technical challenges.
>
> The problem addressed in our paper (linking molecular fragments) is a cornerstone and currently unsolved problem in drug design. We therefore believe that providing a solution to a key biochemical problem using a novel class of ML techniques is important and explicitly falls under the remit of ICLR (“biological applications”). We demonstrate the ideal fit of diffusion models for addressing the molecular linker design problem: it allows us to cover various linker design scenarios that had no existing solution to date such as multi-fragment linking and protein pocket conditioning. **These are unsolved key problems in fragment-based drug discovery and adjacent areas (see Question 3 for examples) that we successfully address.** Besides, our model outperforms other linker design methods on the standard benchmarks.
>
> Regarding ML innovation, indeed, our work has been inspired by and is based on EDM. However, we develop a diffusion model for a new problem, which requires substantial changes in the mechanism. Our main novel contribution is the 3D-conditioning approach for Euclidean diffusion models that is only concurrently proposed in work [1]. As we show in Section 4.1, it comes with the advantage of more easily incorporating equivariance properties compared to the EDM framework. For instance, it removes the need to operate on a center-of-mass-free (CoM-free) system. As a result, the denoising mechanism of DiffLinker is significantly simpler.
>
> Finally, we curate two new benchmarks for molecular linker design and utilize additional metrics for assessing the generated molecules to provide practitioners with a better understanding of capabilities of the linker design methods. These contributions address shortcomings of previous evaluations and extend the range of scenarios in which molecular linker design methods can be tested.
>
> **To sum up, we: (a) provide a method that solves a new biological problem of high relevance that has never been solved before (b) propose a novel mechanism of the 3D-conditioning for euclidean diffusion model which has substantial advantages over naive inpainting strategy (b) curate two new benchmarks.** We believe that these contributions perfectly fit and are more than sufficient for the “biological application” ICLR section.
>
> > Besides, actually diffusion models itself has the capacity for "inpainting" missing parts. As shown in the original diffusion model paper [1], the diffusion model can naturally recover the missing patches of images. This further indicates this "linker design" is a natural and simple usage of existing diffusion models for molecule generation, making the technical contribution limited.
>
> Indeed, inpainting for molecular linker design can easily be implemented using a vanilla equivariant diffusion model (EDM) with a slight tweak of the sampling function. We initially considered this approach, but then realized that our novel mechanism of 3D-conditioning simplifies requirements for E(3)-equivariance (e.g., no need to operate on a CoM-free system). Apart from conceptual advantages, the 3D-conditioning approach also empirically outperforms the inpainting strategy: **it increases the validity of the generated molecules by 25%, and improves RMSD more than two-fold.** We thank the reviewer for bringing up this very relevant point and add Section A.10 where we discuss the inpainting strategy and the results in Table 7.

---

> > ### Author Response · Authors · 2022-11-11
> > **Response to Reviewer TyBL [Part 2]**
> >
> > > I'm also a little doubtful about the experimental setup. From the domain-specific (biological or chemical) perspective, considering we already have several fragments and want to design the linker for them: why for the different disconnected fragments we can know their relative positions and even exact coordinates before generating the linker (as shown in Fig.1). I think for this linker design problem, we should take these fragments' positions as known, which, however, is even treated as rigid bodies in this paper.
> >
> > We thank the reviewer for critically assessing the experimental setup and are happy to share more information on biological relevance of the proposed problem setting. We provide additional detail on **major drug discovery directions in which connecting fragments placed at fixed positions in space is an extremely relevant and sometimes preferred strategy** in appendix A.1. Below we list and briefly discuss main examples:
> >
> > _Fragment-based drug discovery (FBDD)_
> >
> > By analogy with classical drug discovery methods, one of the common strategies in FBDD is to operate on fragments that strongly interact with the target proteins. First, strongly binding fragments are experimentally discovered. At this step, the exact location and orientation of the fragments, in which they bind strongly to the target, is determined. The next step is to find a linker between the fragments that preserves positions and thus the binding strength of the fragments [2]. There have been multiple successful drug discovery works in which the starting point was a crystal structure of a protein with fragments bound to it [2]. To name a few, inhibitors for CK2 [3], LDH-A [4] and Dot1L [5], which are proteins playing crucial roles in progress of various cancers, were designed by linking the fragments that were experimentally observed in a bound state with the corresponding targets.
> >
> > _Proteolysis targeting chimera (PROTAC)_
> >
> > PROTAC is a heterobifunctional small molecule designed for stimulating degradation of a target protein by connecting it to an E3-ligase. PROTACs consist of two ligands joined by a linker: one ligand recruits and binds a target protein while the other recruits and binds E3 ubiquitin ligase [6]. For designing PROTACs, one of possible strategies is to dock two proteins (with ligands bound to them) to explore a favorable conformation of the prospective tertiary complex. This information about the initial docking pose of the proteins and exact positions of bound fragments is further used for designing a linker that will stabilize the whole complex [7,8].
> >
> > _Scaffold hopping_
> >
> > Scaffold hopping is a strategy for designing novel compounds by replacing the central core structure of the known molecule. As shown in [9], various scaffold-hopping strategies rely on the experimental 3D data of the initial compound bound to a target complex: the information about the geometry of the initial bound molecule is important for altering its core with the increase of the binding affinity, potency or selectivity of the whole molecule. In such a case, scaffold-hopping of the bound molecule can be considered as a linking problem of several disconnected fragments with fixed known 3D coordinates.
> >
> > **To sum up, our experimental setup is highly relevant: information about relative positions of fragments and their exact coordinates is of high importance for designing valid and functional linkers.** And thanks to operating on 3D coordinates, DiffLinker can be successfully applied in various drug design strategies listed above and in Appendix A.1.
> >
> > > Addressing  the reviewer’s concern about treating fragments as rigid bodies.
> >
> > We thank the Reviewer for bringing up a very interesting point about a possible extension of our method to treat input fragments as non-rigid bodies and potentially alter their initial positions in order to obtain a more favorable configuration of the resulting molecule. Even though such an extension opens up new opportunities in designing novel compounds, it also puts substantial challenges in preserving high strength of interaction between the fragments and the target protein upon changing their positions. Therefore, taking into account both potency and complexity of this problem setting, we do believe that such an approach itself can be considered as a substantial contribution to the field, but we leave it for future work.

---

> > > ### Author Response · Authors · 2022-11-11
> > > **Response to Reviewer TyBL [Part 3]**
> > >
> > > > I'm also concerned about the "reflection" equivariance imposed on the proposed model. According to my knowledge, the energy of reflected biostructures should be different, which means they shouldn't have reflection symmetry. However, this is not the case for the proposed model.
> > >
> > > Indeed, chirality plays an important role in intermolecular interactions. However, the energy of the molecule itself (in void) is the same as the energy of any its enantiomer [10]. **Our method completely takes it into account**: if we consider reflections of the whole molecule (fragments + linker), the likelihood of any enantiomer will be the same because our model is equivariant: $p(\boldsymbol{x}|\boldsymbol{u}) = p(\boldsymbol{R}\boldsymbol{x}|\boldsymbol{R}\boldsymbol{u})$ for any $\boldsymbol{R}\in O(3)$.
> > >
> > > Talking about molecular interactions, we want to (a) distinguish interactions between a protein pocket and different small molecule enantiomers and (b) distinguish molecules that are made of reflection-symmetric fragments and different linker enantiomers. Both cases fall under the following requirement: $p(\boldsymbol{R}\boldsymbol{x}|\boldsymbol{u})\neq p(\boldsymbol{x}|\boldsymbol{u})$. By default, this condition is true for chiral linkers, and implementation-wise, our method does not have components that would violate this requirement.
> > >
> > > > [minor point] Considering this is an application paper, actually I'm not sure whether the evaluation itself is sound. For the dataset part, the data is obtained by manually subtracting fragments from molecular structure datasets. This makes me feel like the benchmark itself is human-designed and far away from real-world challenges. I understand this setup may be already adopted in a few existing benchmarks, but this doesn't mean we can just follow them despite the limitation in existing benchmarks. Maybe better and more realistic benchmarks can also be introduced.
> > >
> > > Benchmarks were created using two different methodologies of splitting molecules into fragments [11, 12]. Both algorithms constitute sets of rules for breaking covalent bonds in a molecule such that the most stable compounds remain unchanged. This approach is in line with the idea of fragment-based drug discovery, where the starting points are chemically-stable small compounds. Therefore, the benchmarks we used in our work are close to the real FBDD applications, and a lot of fragments we have in our datasets can be found in libraries of commercially-available compounds. Besides, the same methodologies that we used for constructing datasets are applied in scaffold hopping, where the central core of the molecule (in our terminology, “linker”) is modified. All the same ideas about chemical stability of certain parts of molecules are used for defining the core compounds to be altered [13]. Therefore, we would like to emphasize the relevance of the proposed benchmarks for the applications.
> > >
> > > ---
> > >
> > > We hope that our answers have clarified the points raised and increased the reviewer's confidence in our work. If the reviewer has other doubts, we would be happy to engage in a follow-up discussion. If not, we would be glad to have the given score reconsidered.
> > >
> > > **References:**
> > >
> > > [1] Luo S. et al. Antigen-specific antibody design and optimization with diffusion-based generative models, 2022
> > >
> > > [2] Bancet A. et al. Fragment linking strategies for structure-based drug design, 2020
> > >
> > > [3] De Fusco C. et al. A fragment-based approach leading to the discovery of a novel binding site and the selective CK2 inhibitor CAM4066, 2017
> > >
> > > [4] Kohlmann A. et al. Fragment growing and linking lead to novel nanomolar lactate dehydrogenase inhibitors, 2013
> > >
> > > [5] Möbitz H. et al. Discovery of potent, selective, and structurally novel Dot1L inhibitors by a fragment linking approach, 2017
> > >
> > > [6] Békés M., Langley D. R., Crews C. M. PROTAC targeted protein degraders: the past is prologue, 2022
> > >
> > > [7] Bai N. et al. Rationalizing PROTAC-mediated ternary complex formation using Rosetta, 2021
> > >
> > > [8] Farnaby W. et al. BAF complex vulnerabilities in cancer demonstrated via structure-based PROTAC design, 2019
> > >
> > > [9] Sun H., Tawa G., Wallqvist A. Classification of scaffold-hopping approaches, 2012
> > >
> > > [10] Pályi G. et al. Progress in biological chirality, 2004.
> > >
> > > [11] Hussain J., Rea C. Computationally efficient algorithm to identify matched molecular pairs (MMPs) in large data sets, 2010
> > >
> > > [12] Degen J. et al. On the Art of Compiling and Using 'Drug-Like' Chemical Fragment Spaces, 2008
> > >
> > > [13] Vainio M. J. et al. Scaffold hopping by fragment replacement, 2013

---

### Official Review · Reviewer_3ysG · 2022-10-24

**Confidence:** 3
**Correctness:** 4
**Technical Novelty And Significance:** 3
**Empirical Novelty And Significance:** 4
**Recommendation:** 8

**Clarity, Quality, Novelty And Reproducibility:**

Clarity (high): the paper is written clearly in both the method and experiment sections.
Quality (high): the paper has high-quality figures and results to facilitate the understanding of readers.
Novelty (high to fair): while the ideas in diffusion models have been explored in prior work such as GeoDiff, the application to molecular linker design is novel.
Reproducibility (high): code is provided to encourage reproducible results, from code to datasets. Reading the readme, it seems that downstream users should be able to run this code.

**Strength And Weaknesses:**

Strength:
- Description of the method is generally clear.
- This is a novel application of diffusion models on 3d molecule linker design.
- Code is provided to encourage reproducibility.
- A thorough ablation study is given to DiffLinker with regards to given anchors and sampled size.

Weakness:
- The method is described specifically for DDPM, so it inherits all the efficiency issues of DDPM as well.
- It might help to give some context of some domain-specific settings, especially protein pockets and their representations.

**Summary Of The Paper:**

The paper presents a diffusion model approach for predicting molecular linkers between disconnected fragments. This is a two stage process: a model first predicts linker size, and then a diffusion model produces the linker results. Empirical results demonstrates the effectiveness of the proposed method compared with baseline methods.

**Summary Of The Review:**

The paper presents an interesting application of diffusion models to molecule linker design. While the method itself is straightforward, the method demonstrates the utility of diffusion models over autoregressive methods. The idea of the two-stage process is also interesting and could be useful in other applications.

---

> ### Author Response · Authors · 2022-11-11
> **Response to Reviewer 3ysG**
>
> We thank the reviewer for the positive feedback and we are glad that the reviewer finds our wok to be “a novel application of diffusion models on 3d molecule linker design” and appreciates “a thorough ablation study is given to DiffLinker with regards to given anchors and sampled size.” We thank the reviewer for raising the questions about general efficiency issues of DDPMs and are happy to clarify the way how we represent protein pockets.
>
> > The method is described specifically for DDPM, so it inherits all the efficiency issues of DDPM as well.
>
> It is true that while denoising diffusion models are trained very efficiently and are remarkably stable (compared to GANs or normalizing flows), sampling with them is slower. However, as it was shown in [1], the number of denoising steps in sampling can be significantly reduced with little to no performance degradation. In Section A.4, we have added a discussion of sampling with DiffLinker and conducted additional experiments comparing the performance and sampling time of DiffLinker with different numbers of denoising steps (T, T/2, T/5, T/10, T/20, T/50, T/100) in sampling. As shown in (new) Figure 3, **DiffLinker is robust to reducing the number of denoising steps in sampling resulting in 10-fold gain in sampling speed without any performance degradation**. Besides, we provide an additional Table 3 where we report DiffLinker sampling time (with T=500) for all the datasets.
>
> > It might help to give some context of some domain-specific settings, especially protein pockets and their representations.
>
> First of all, to construct a pocket, we selected all the protein residues that have at least one atom that is closer than 6A to any atom of the ligand. Having the residues selected, we considered two different scales of the protein pocket representation: (1) point cloud of all atoms belonging to the pocket residues and (2) only backbone atoms CA, C, N, O. For each pocket atom, we assigned its one-hot encoded atom type as a feature vector. In our framework, conditioning on the pocket does not differ from conditioning on fragments – we just have more atoms in the graph. There are only two minor changes we have introduced to the graph construction procedure: (a) additional binary flag indicating that an atom belongs to the protein and (b) instead of considering a fully-connected graph, we assigned edges only between atoms that are closer than 4A (because graphs with protein pockets are much larger). We considered two different representations to demonstrate how the amount of the information about surrounding atoms affect the number of clashes between the generated linkers and protein pockets (Figure 2). Details about pocket construction and processing are provided in Sections 4.4 and 5.1.
>
> **References:**
>
> [1] Improved denoising diffusion probabilistic models, Nichol & Dhariwal 2021

---

> > ### Comment · Reviewer_3ysG · 2022-11-14
> > **Thanks for the response**
> >
> > Thanks for the detailed response and the revision.
> >
> > While this is not very important to my rating, I am surprised to see how the Improved DDPM for 50 steps performs similarly to that of 500 steps. Perhaps with even better samplers such as DDIM and DEIS the number of iterations could be further decreased.

---

### Official Review · Reviewer_LoJe · 2022-10-25

**Confidence:** 2
**Correctness:** 4
**Technical Novelty And Significance:** 3
**Empirical Novelty And Significance:** 3
**Recommendation:** 6

**Clarity, Quality, Novelty And Reproducibility:**

The paper is in good quality and reproducibility is good with the provided code.

**Strength And Weaknesses:**

Strength:
1. The paper is written very clearly with each step and model overview.
2. The results on linker design is very impressive and pioneering the study in the field.

Weakness:
1. It is not very clear on adopting diffusion process in conditional linker design. What's the advantage of VAE-based or Normalizing Flow-based algorithms?
2. What's the limitation and common failures in the proposed generation process?

**Summary Of The Paper:**

This paper proposes DiffLinker, which is a diffusion model for molecular linker design that can link an arbitrary amount of fragments. It features an E(3)-equivariant model in the diffusion process and 3-D conditional distribution on input fragments and optional protein pocket. The result shows superior performance on various real-world applications including a newly proposed one in the paper.

**Summary Of The Review:**

This paper tackles conditional generation in molecular design using recent diffusion model and equivariant graph neural networks. The generation quality is impressive compared with existing work.

---

> ### Author Response · Authors · 2022-11-11
> **Response to Reviewer LoJe**
>
> We are glad that the Reviewer found our work important for the progress of the fragment-based drug design field. We thank the Reviewer for the interesting questions, which we address below, providing a comparison of different generative models and discussing advantages and weaknesses of our method.
>
> > It is not very clear on adopting diffusion process in conditional linker design. What's the advantage of VAE-based or Normalizing Flow-based algorithms?
>
> While the implementations of VAEs, normalizing flows and diffusion models are very different, it is not trivial to theoretically disambiguate between these models. Here are a few similarities and differences:
>
> *Comparison with VAEs:*
> * The graphical model of diffusion models is the same as the graphical model for hierarchical VAEs – this implies in particular that the ELBO computation is similar.
> * A diffusion model can therefore be seen as a hierarchical VAE where the encoder is fixed (it simply adds noise to the data). One advantage of this formulation is that the latent space is (close to) Gaussian by design, and that only the decoder needs to be learned.
> * Contrary to hierarchical VAEs, diffusion models are trained in parallel on the different time steps.
>
> *Comparison with normalizing flows:*
> * Diffusion models and normalizing flows both map a normal distribution at time t=0 to a model for the data distribution at time t=1.
> * Diffusion models do not use the change of variable formula (they optimize an ELBO and not directly the model likelihood). They are therefore not constrained as some normalizing flows (but not all) are.
> * The main difference is again computational: normalizing flows are typically trained by applying the neural network iteratively, while diffusion models are trained in parallel over the different time steps. Applying the network iteratively means having to deal with various stability issues: for example, the authors of [1] report a training time of two weeks on the QM9 dataset, despite this dataset being relatively simple. Diffusion models circumvent this issue and are therefore much easier to train.
> * In fact, it is possible to cast normalizing flows as diffusion models [2], but the resulting algorithm is much closer to a diffusion model than to a standard normalizing flow.
>
> Overall, there is extensive evidence on images [3], video [4], audio [5] and graphs [6] that diffusion models tend to strongly outperform VAEs and normalizing flows. We believe that it is not really due to a fundamental difference in the mathematical formulation (diffusion models only use Bayes formula and an ELBO computation), but more to a diffusion model  algorithm that is numerically stable (the neural network is not applied recursively) and prone to parallelization.
>
> In the scope of the molecular linker design problem, for which two VAE-based methods were previously proposed [7,8], **we especially emphasize the advantage of the one-shot linker generation process over the autoregressive generation process** that has been employed in previous works. As explained in Section 5.3, DiffLinker is naturally capable of connecting arbitrary numbers of fragments without partial loss of the context (which is not the case for the other methods). Therefore, taking into account the recent success of diffusion models in various domains and the particular advantage of the one-shot generation in our task, we believe (and demonstrate) that diffusion model is a perfect candidate for applying in molecular linker design.
>
> > What's the limitation and common failures in the proposed generation process?
>
> First, as well as other diffusion models, DiffLinker tends to be slower than other generative methods at the inference time. Even so, we should emphasize that the sampling time of DiffLinker is not a limitation for applying it in practice as it is robust to significant reduction of the number of denoising steps in sampling. We have added the discussion of sampling with DiffLinker in Section A.4. Besides, we added Figure 3 that compares performance of DiffLinker with different numbers of denoising steps, and Table 3 that provides DiffLinker sampling time for all the datasets.
>
> Besides, our model outputs 3D coordinates that we later use to derive the covalent bonds. While we observed that our model successfully learns bonding rules by itself, it would be interesting to have in the future a model that generates both the 3D coordinates and the graph structure in an end-to-end fashion.

---

> > ### Author Response · Authors · 2022-11-11
> > **References**
> >
> > [1] Garcia V. et al. E(n) equivariant normalizing flows, 2021
> >
> > [2] Lipman Y. et al.  Flow Matching for Generative Modeling, 2022
> >
> > [3] Dhariwal & Nichol  Diffusion Models Beat GANs on Image Synthesis, 2021
> >
> > [4] Ho et al. Video diffusion models, 2022
> >
> > [5] Kong et al. Diffwave: a versatile diffusion model for audio synthesis, 2020
> >
> > [6] Vignac et al. DiGress: discrete denoising diffusion for graph generation, 2022
> >
> > [7] Imrie F. et al. Deep generative models for 3D linker design, 2020
> >
> > [8] Huang Y. et al. 3DLinker: An E (3) Equivariant Variational Autoencoder for Molecular Linker Design, 2022

---

### Author Response · Authors · 2022-11-11
**Summary of responses and changes**

We thank all the reviewers for the time they spent on reviewing our work. We are happy to see a lot of positive feedback from the reviewers. We did our best to respond to all individual questions in the direct comments and made all the necessary modifications in the submission text. Below we summarize the changes we made to address reviewers’ comments (in the manuscript all the changes are colored in blue).

1. Addressing the comment about the relevance of DiffLinker problem setting, we added Section A.1 that lists major drug discovery directions (fragment-based drug design, PROTACs, scaffold hopping) in which **connecting fragments placed at fixed positions in space is a highly relevant and sometimes preferred strategy**.
2. Responding to the questions raised by two reviewers, we added Section A.10 discussing applicability of the “inpainting” approach for molecular linker design. To demonstrate that the proposed 3D-conditioning mechanism outperforms inpainting strategy, we provide a new Table 8 with quantitative comparison of two approaches. According to the results, **the 3D-conditioning mechanism increases the validity of the generated molecules by 25%, and improves RMSD more than two-fold**.
3. To answer the questions about efficiency of DiffLinker, we discussed the sampling process in Section A.4  and added Table 3 with sampling time for different datasets. Besides, we conducted additional experiments on reducing the number of sampling steps, summarized the results in Figure 3, and demonstrated that **DiffLinker is robust to 10-fold reduction of the number of denoising steps in sampling**.
4. We added Table 7 to report QED, SA and #Rings scores on the molecules from the train, validation and test sets.
5. We updated Figure 2 fixing molecular 2D-representations and providing molecules with higher synthetic accessibility scores.
6. We updated Tables 1, 2, 8 and 9 (GEOM-related numbers) due to the bug found in the test set (one test molecule was represented in the training set). We note that there is no significant change in the results. Besides, in new Table 10 we provide additional results for the modified Pocket test set (with 17 molecules excluded).

We hope that our submission modifications and answers will clarify the raised points and increase the reviewers’ confidence in our work. We are happy to engage in the follow-up discussion and welcome any additional comments.

---

### Decision · Program_Chairs · 2023-01-20

**Decision:**

Reject

**Justification For Why Not Higher Score:**

Major concerns are that the technical novelty and advances are limited, and the problem setting has limited impacts in reality, as the actual application context is not clear.

**Justification For Why Not Lower Score:**

NA

**Metareview: Summary, Strengths And Weaknesses:**

This paper studies the problem of generating a linker to connect two fragments of molecules. The problem setting was first proposed in the 3Dlinker paper at ICML 2022, which used VAE. This paper proposes to solve the same problem with diffusion models. A strength of this paper is that, by using diffusion models, the performance has been improved in many cases as compared to VAE in 3Dlinker. Major concerns are that the technical novelty and advances are limited, and the problem setting has limited impacts in reality, as the actual application context is not clear.

**Summary Of Ac-Reviewer Meeting:**

An AC-reviewer meeting was hold for this paper. The reviewers who are positive about this paper mainly value its practical significance and potential impacts, while they also admit the technical novelty and advances are limited. The reviewers who are not supportive to this work are mainly concerned with technical novelty and advances and limited impacts in reality, as the actual application context is not clear. One of the positive reviewers downgraded his/her rating after the conference.